# Interpretable and context-free deconvolution of multi-scale whole transcriptomic data with UniCell deconvolve

Daniel Charytonowicz[1], Rachel Brody[2] & Robert Sebra [1,3,4] ✉

We introduce UniCell: Deconvolve Base (UCDBase), a pre-trained, interpretable, deep learning model to deconvolve cell type fractions and predict cell identity across Spatial, bulk-RNA-Seq, and scRNA-Seq datasets without contextualized reference data. UCD is trained on 10 million pseudo-mixtures from a fully-integrated scRNA-Seq training database comprising over 28 million annotated single cells spanning 840 unique cell types from 898 studies. We show that our UCDBase and transfer-learning models achieve comparable or superior performance on in-silico mixture deconvolution to existing, reference-based, state-of-the-art methods. Feature attribute analysis uncovers gene signatures associated with cell-type specific inflammatory-fibrotic responses in ischemic kidney injury, discerns cancer subtypes, and accurately deconvolves tumor microenvironments. UCD identifies pathologic changes in cell fractions among bulk-RNA-Seq data for several disease states. Applied to lung cancer scRNA-Seq data, UCD annotates and distinguishes normal from cancerous cells. Overall, UCD enhances transcriptomic data analysis, aiding in assessment of cellular and spatial context.

The ability to measure expression of the coding genome has revolutionized the study of human disease[1]. Recently, the appreciation of inter-patient cellular heterogeneity has led to methods such as single-cell RNA Sequencing (scRNA-Seq) being introduced to increase study resolution[2]. There is now interest in measuring the influence of spatial cellular organization on pathophysiology, which is being accomplished through Spatial Transcriptomics (ST). Broadly, ST platforms can be divided into two categories. Targeted, high-resolution approaches such as MERFISH[3], split-FISH[4], or OligoFISSEQ[5] can profile tens to hundreds of genes using variations of nucleic-acid hybridization techniques at the subcellular level. Alternatively, whole-transcriptome, lower-resolution approaches such as Slide-Seq[6], Visium[7], DBiT-seq[8], or Stereo-seq[9] function via spatial-aware RNA capture and sequencing. The unbiased nature of whole-transcriptome approaches makes them appealing for early-stage discovery and hypothesis-generation.

Resolution of whole-transcriptome spatial platforms varies, ranging from 10 um for Slide-Seq to 55 um for Visium. While the density of capture arrays is increasing, spatial capture spots nevertheless contain RNA content eluted from several single cells. Differences in gene expression are driven in-part by varying cell type mixtures and levels of individual cell transcript expression. As such, it is essential to "deconvolve" cell type fractions for each spot to improve interpretability and analysis of differential gene expression patterns. Multiple machine learning methods addressing cellular deconvolution have been introduced. Earlier approaches focusing on bulk-RNA-Seq include methods such as DSA[10], MuSiC[11], CIBERSORT/CIBERSORTx[12,13], Scaden[14], DeconRNASeq[15], and SCDC[16]. The emergence of ST has ushered in several next generation deconvolution algorithms, notably Cell2Location[17], SPOTLight[18], Stereoscope[19], SpatialDWLS[20], DSTG[21], STDeconvolve[22], and RCTD[23].

[1]Department of Genetics and Genomic Sciences, Icahn School of Medicine at Mount Sinai, New York, NY, USA. [2]Department of Pathology, Molecular and Cell-Based Medicine, Icahn School of Medicine at Mount Sinai, New York, NY, USA. [3]Icahn Genomics Institute, New York, NY, USA. [4]Black Family Stem Cell Institute, New York, NY, USA. ✉e-mail: robert.sebra@mssm.edu

A significant limitation of most approaches is the requirement for a reference profile of cell type expression. Meta-analyses of RNA-seq deconvolution algorithms have shown that choice of reference is more important than methodology in determining deconvolution performance[24]. The choice of cell types to include in a reference is not always apparent, and collecting matched samples for reference generation is not always possible. Furthermore, the use of general scRNA-Seq "atlases" as references may not be appropriate when transcriptional differences due to experimental or disease-related factors confound cell type expression patterns. It has been suggested that the integration of numerous studies with varying experimental conditions and technical platforms can improve the robustness and generalization of deconvolutions[25].

To address these challenges, we introduce UniCell: Deconvolve Base (UCDBase), a pre-trained, context-free, deep learning foundation model for universal cell type deconvolution. UCDBase is trained using 10 million pseudobulk RNA mixtures generated from the world's largest fully integrated scRNA-Seq database, comprising 28 million fully-annotated single cells representing 840 cell types collected from 899 uniformly preprocessed, validated, and published single-cell datasets. First, we describe the collection and integration strategy used to build training data for UCD, and then detail the architecture of our model.

We demonstrate how UCDBase performance compares favorably to existing reference-based approaches, with feature attribute analysis enabling orthogonal validation of predictions by associating gene expression with particular cell types. UCDBase can also be leveraged as a global cell type feature extractor for transfer learning given user-specified cell signatures, facilitating the rapid deployment of context-specific deconvolution "UCDSelect" models.

We highlight UCDBase's ability to deconvolve changes to immune and stromal cell infiltrates in response to ischemic kidney injury, associating differentially active stress response genes to kidney epithelial cell types. Next, UCDBase applied to bulk-RNA-Seq data pinpoints specific losses in pancreatic beta cell and oligodendrocyte fractions in type 2 diabetes and multiple sclerosis, respectively. UCDBase also accurately differentiates between cancer subtypes across bulk, spatial and single-cell data. Lastly, UCDBase is used to annotate primary human lung cancer data, providing marker genes to corroborate predictions, and distinguishes normal from cancerous epithelial cells.

## Results

### Single-cell RNA-Seq simulated mixture benchmarking

We compared actual and predicted cell type fractions across simulated mixtures for our three benchmarking datasets comprising PBMC, Lung, and Retina tissues (see Fig. 1a, c, and e). For each mixture set, we compared actual and predicted cell type fractions across 500 simulated mixtures (see Fig. 1b, d, and f). To better evaluate the performance of UCDSelect, we performed an ablation study whereby transfer learning performance of UCDBase embeddings alone was compared with conventional gene feature extraction alone, as well as combined.

For PBMCs, our pre-trained UCDBase model obtained strong concordance correlation coefficient (CCC) values of 0.816 averaged across the eight cell types identified in our dataset, while UCDSelect achieved CCC of 0.864, 0.921 and 0.92 for deconvolution utilizing gene features only, embeddings only, and both sources, respectively. UCDBase performed comparably with current State of the Art methods such as Cell2Location (C2L) (see Fig. 1b top), despite the fact that C2L and competing algorithms were trained to exclusively consider the deconvolution of PBMCs. We note that in the PBMC task, the cell type categories used for comparison are distinct and well-defined, indicating that the corresponding cell types found in UCDBase training dataset are likely to be well-aligned with the labels assigned for this task. UCDSelect exhibited superior performance in this benchmarking task compared with all competing methods.

Results seen in Lung and Retina data highlight the importance of accounting for mismatch between UCDBase and target cell type annotations, and the relevance of UCDSelect as a transfer learning extension of UCDBase. We show that preliminary results indicated average concordance (CCC = 0.524 for Retina, CCC = 0.532 for Lung) with high variance when directly comparing annotated cell types from reference data with the corresponding cell types found in UCDBase's 840 cell type output. We investigated these discrepancies in Supplementary Fig. 8, where we identified cell types with low initial concordance measurements in both Lung and Retina datasets (see Supplementary Fig. 8a, c). We select three low-performing cell types and performed cross-correlation with output vectors of all 840 UCDBase cell types, and plot pearson correlation between the ground-truth labeled cell type and top 16 highest correlated UCDBase outputs (see Supplementary Fig. 8b, d).The results strongly illustrate that UCDBase correctly identifies cellular state identity, albeit the annotation matched within UCDBase does not always perfectly align with those in the target dataset. For example in our lung mixture dataset, "endothelial cells", which show a direct label matched correlation of effectively zero, are identified by UCDBase as correlating most closely with "lung endothelial cells" (pearson's $R = 0.851$). Similar patterns are seen among other examined cell types, supporting the notion that UCDBase is correctly identifying cell types, however label mismatches make it difficult to discern true accuracy when working with benchmarking datasets relying on potentially flawed, user-defined cell types as ground truth labels. It further highlights the importance of detailed interpretation when analyzing the results of a global pre-trained deconvolution model.

UCDSelect however, represents a natural extension of UCDBase and a solution to the complexity of label mismatch. By aligning UCDBase's feature vectors to a user-specified reference signature, we are effectively able to guide UCD to a solution within the parameter space defined by the user. For the Lung benchmark, UCDSelect achieves average CCC values of 0.832, 0.861, and 0.883 for features, embeddings, and both sources, respectively. The Retina benchmark saw average CCC values for UCDselect of 0.93, 0.97, and 0.972 for features, embeddings, and both sources, respectively. The strong performance on the Retina benchmark is unsurprising, given that unlike the PBMC and Lung datasets which featured mixture and reference data derived from different studies, the paired Retina reference and mixture data sources are both derived from two samples from the same study, which likely minimizes the batch and/or experimental related differences between common cell types in these samples.

### Downsampled spatial transcriptomic data

We measured the performance of UCDBase and UCDSelect in deconvolution of downsampled *mouse* hippocampal Slide-SeqV2 spatial transcriptomic data (Fig. 1g). We highlight strong visual concordance between three representative ground truth hippocampal cell type annotations and UCDBase / UCDSelect predictions in Fig. 1h. To quantify performance, we deconvolve downsampled mixtures using several comparator methods developed for spatial transcriptomics, and show that UCDSelect exhibits comparable deconvolution performance relative to state-of-the-art reference-based approaches, with average CCC values of 0.511, 0.532, and 0.561 for features, embeddings, and both sources, respectively (Fig. 1i). Stereoscope and Tangram showed the most consistent performance on this dataset, with average CCC of 0.616 and 0.588, respectively.

### Bulk RNA-Seq benchmarking

We compared the performance of UCDBase and UCDSelect in deconvolution of gold-standard bulk RNA-Seq mixture and reference profiles developed for the community DREAM bulk RNA-Seq

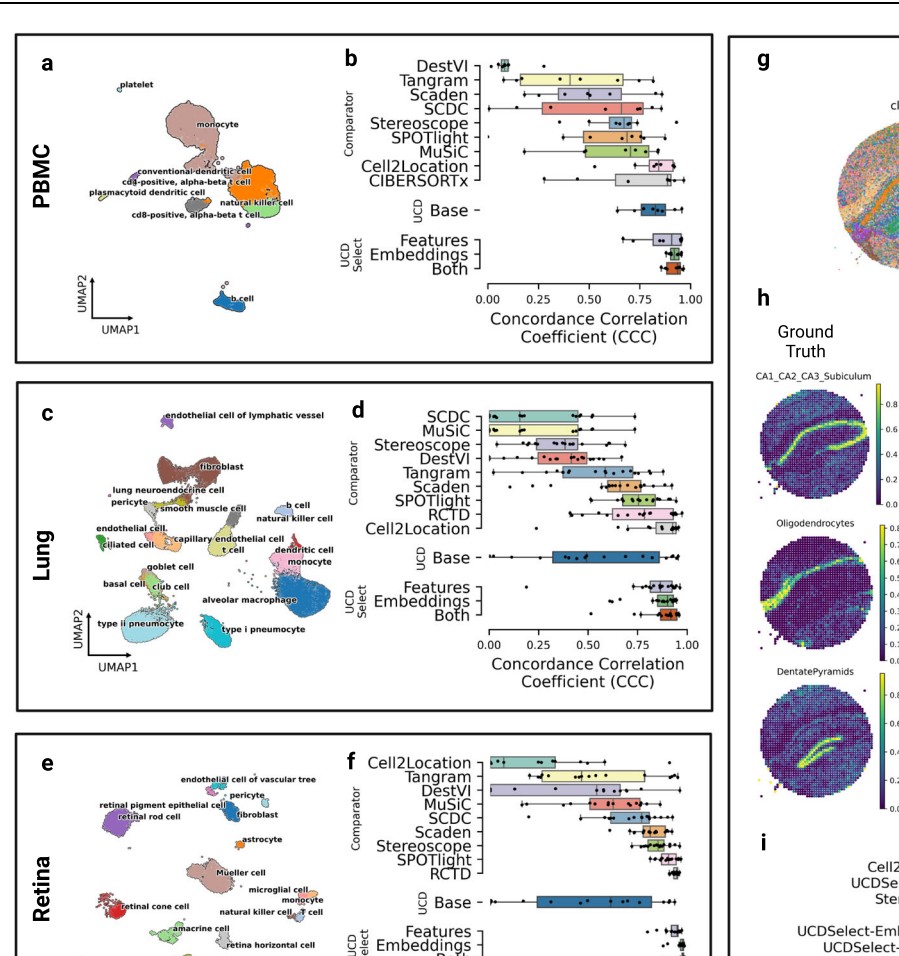

**Fig. 1 | Benchmarking UniCell deconvolution performance across tissue types.**
**a** UMAP visualization of *human* peripheral blood mononuclear cell (PBMC) single cells used to generate pseudobulk mixtures for deconvolution benchmarking, annotated by cell type. **b** Box plots of deconvolution performance for each cell type (*n* = 8) in the PBMC dataset, stratified by method (*y*-axis), as measured by concordance correlation coefficient (*x*-axis). **c** UMAP visualization of *human* lung tissue single cells used to generate pseudobulk mixtures for deconvolution benchmarking, annotated by cell type. **d** Box plots of deconvolution performance for each cell type (*n* = 19) in the lung dataset, stratified by method (*y*-axis), as measured by concordance correlation coefficient (*x*-axis). **e** UMAP visualization of *human* retina periphery single cells used to generate pseudobulk mixtures for deconvolution benchmarking, annotated by cell type. **f** Box plots of deconvolution performance for each cell type (*n* = 17) in the retina dataset, stratified by method (*y*-axis), as measured by concordance correlation coefficient (*x*-axis). **g** Spatial profile of *murine* hippocampal formation profiled using Slide-SeqV2 colored by individual

cell type. **h** Spatial heatmaps representing a downsampled hippocampal dataset, where each spot represents the average gene expression profile obtained from multiple individual cells in close spatial proximity. The first column illustrates the ground truth fractions of three representative cell types comprising the downsampled spatial spots (with the scale ranging from 0 to 1 representing 0% to 100% of cells in that downsampled spatial spot corresponding to a given cell type). The middle column denotes cell fraction predictions for matched or related cell types given by UCD Base. The rightmost column denotes cell type predictions made by UCD Select trained on individual cell profiles from the source dataset. **i** Box plots of deconvolution performance for each cell type (*n* = 14) in the hippocampal dataset, stratified by method (*y*-axis), as measured by concordance correlation coefficient (*x*-axis). For boxplots in **b**, **d**, **f** and **i**, the center line, box limits and box whiskers correspond to the median, first and third quartiles, and the 1.5x interquartile range, respectively. Individual data points are superimposed over each boxplot.

deconvolution challenge with respect to the results obtained by submitted competitor methods (see Supplementary Fig. 9)[26]. UCDBase achieved a mean score (measured by pearson's R) of 0.68 when deconvolving 96 cell mixtures, placing it in the top half of solutions. In contrast, UCDSelect achieved mean Pearson's R scores across 11 compared cell subtypes of 0.793, 0.892, and 0.903 respectively, scoring considerably higher than competing approaches.

### Hyperparameter sensitivity analysis
UCDBase and UCDSelect were found to be robust to changes in mixture hyperparameters (see Supplementary Fig. 10) across our three synthetic mixture datasets. We saw a minimal linear decrease in mean

performance as sample complexity (i.e. number of unique cell types) increased (see Supplementary Fig. 10b, e and h). Model performance was found to be consistent while varying the number of cells used to generate each mixture, with a slight reduction for lower total mixture cell counts, which we believe is caused by increased signal-to-noise ratio (see Supplementary Fig. 10c, f and i).

When perturbing gene dropout, we found that significant performance reductions were seen only after >80% of expressed genes in the benchmarking mixture samples were removed as inputs. This robustness to dropout suggests that UCDbase leverages nonlinear combinations of gene sets as the basis of cell type fraction predictions, and is resilient to the noise seen in transcriptomic data, especially at

lower read depth. It nevertheless suggested that the current UCDBase architecture may not be appropriately tuned for use with technologies profiling smaller numbers of genes. To validate this important distinction, we obtained mixture and reference signatures generated by Li. et al. 2022 derived from the mouse visual cortex using the in-situ STARmap spatial transcriptomic technology (see Supplementary Fig. 11a, b)[27]. With an input of just 881 genes, we reasoned that UCDBase performance would be limited by such a degree of sparsity (~97%) relative to the whole transcriptome input space it was trained on. Unsurprisingly, we see that UCDSelect achieves only modest deconvolution performance (CCC = 0.658, 0.567, and 0.64 for features, embeddings and both sources, respectively). Notably, results indicate that in this scenario, gene expression features, as opposed to UCDBase extracted embeddings, provide superior deconvolution performance (see Supplementary Fig. 11c). We therefore suggest to users that UCD be utilized primarily in cases where whole transcriptome data is available so as to maximize accuracy and performance.

### Training data composition and sensitivity across selected technology platforms

As UCDBase was trained using a comprehensive collection of studies collected using a range of technology platforms, from a range of sources including different species, we sought to better understand the composition of our training dataset, and assess the impact, if any, on UCDBase deconvolution accuracy. Utilizing keyword extraction on metadata available for each collected project, we assessed the most likely technology platform used to generate the collected dataset. For instances where multiple technologies were identified, the most common and/or first occurring keyword was assigned as a label for that study. A similar keyword approach was used to detect species for each dataset, with semicolons denoting multiple potential species identified for a given study. We report on the results of this technology and organism assessment in Supplementary Fig. 12. We estimate that ~66.4% of our collected cells were generated using a version of the 10X Genomics Chromium platform (see Supplementary Fig. 12a). 95.1% of cells in our database are estimated to derive from short-read sequencing data, with just 4.9% coming from technologies such as Smart-Seq. UCDBase and UCDSelect performance was then benchmarked for deconvolution of synthetic mixtures generated from PBMCs derived using a number of scRNA-Seq technologies from a study conducted by Ding et al. 2019[28]. We highlight the performance of this comparison in Supplementary Fig. 13, where we demonstrate that both UCDBase and UCDSelect show comparable performance in deconvolution accuracy across multiple platform technologies, including Smart-Seq long-read data.

Looking at species origins, *human* derived data made up the majority of cells in our database at 43.2%, with 22.1% coming from mice, and 33.1% of cells coming from datasets where both *Human* and *Mouse* keywords were found. By and large the vast majority of single cell data in our training dataset (98.4%) is derived from either *Human* or *Mouse* sources, which represent the most common species subject to single cell analysis (see Supplementary Fig. 12b). Across all matched cell types, the average correlation between gene expression across *mouse* and *human* data was found to be moderately positive (pearsonr $r = 0.46$) (see Supplementary Table 4). Given the potential discrepancies in gene expression between species, we therefore suggest that users bear in mind the species of origin when utilizing UCDBase given the species composition underpinning its training dataset.

### Characterization of pathophysiologic cell type aberrations in ischemic kidney injury

Kidney ischemia reperfusion injury (IRI) describes the oxidative stress and inflammatory damage induced by revascularization following a loss of blood flow and oxygen to cells of the renal system[29]. IRI is a common perioperative complication occurring during major trauma,

shock, sepsis, or transplant, and understanding the pathophysiologic changes it induces is critical in developing strategies to mitigate its long term impacts[30]. Using temporal spatial transcriptomics data of coronal kidney tissue sections collected from a *mouse* bilateral renal IRI model, developed by Dixon et al. 2022[31], we leveraged UCDBase to explore changes in kidney cell fractions associated with progressive IRI damage (see Fig. 2a).

We began by examining deconvolution results in the context of normal control tissue in Fig. 2c, comparing it with expected cellular organization as summarized in Fig. 2b. UCDBase identified spatial distributions of proximal (PCT) and distal convoluted tubule epithelial cells localizing correctly to the outer cortex zone of the kidney. The thick-ascending limb of the loop of henle (TAL/LOH) was localized to the inner-renal medulla, while cells of the collecting duct (CD) were identified to be distributed across the renal cortex with increased abundance in the medulla, as they coalesce into the renal calyx. Intercalated cells (IC) were identified mainly along the boundary zone of the outer medulla, consistent with IC preferential localization in the earlier sections of the CD[32]. UCDBase also predicted "brush cells" in the outer medullary zone, which we suspect correspond to the S3 straight segment of the PCT based on identified gene attributes (see Supplementary Table 2). This is unsurprising, as the morphology of PCT cells is brush-border like, and the S3 segment displays the least degree of functional differentiation[33]. Specific genes UCDBase associated with all renal cell types were contrasted with established literature and are detailed in Supplementary Table 2.

We next examined changes in absolute cell type fractions predicted to occur following IRI. The overall composition and spatial organization of major kidney cell types remained unchanged (see Fig. 2c-center & right). Increases in t cell, suppressor macrophage, and fibroblast content became apparent as early as 2 days post-IRI compared with control, peaking at the 6 week timepoint (see Fig. 2d, e).

A notable gene attributed to t cells was *Ccr7*. It has been shown that *Ccr7* + t cells mediate kidney injury during transplant allograft rejection, suggesting a similar role in IRI[34]. Suppressor M2-like macrophages promote kidney repair after acute IRI by modulating innate immunity[30].

Fibroblast infiltrate at the 6 week timepoint (see Fig. 2h-left) was, to our surprise, associated with complement factor-H (*cfh*) expression. The authors of the original study explicitly noted the inability to establish a link between *cfh* and fibroblasts from Visium data alone, and verified its selective expression among kidney fibroblasts using an independent single-nucleus RNA-Seq dataset[31].

While the canonical PCT marker *Slc34a1* remains a consistent attribute of PCT cells across time points (see Fig. 2g), we see evidence of secondary markers overexpressed following injury, suggesting temporal physiologic changes to PCT cell function. The metabolic waste efflux pump *Abbc2* has been shown to be overexpressed after acute renal IRI in mice[35], and exhibits increased attribution for PCT cells at 12 h post-injury, suggesting overexpression and increased PCT stress[35].

Together, UCDBase enables us to rapidly paint a comprehensive picture of physiological changes underpinning the kidneys' response to IRI. Through cell type deconvolution in addition to feature attribute analysis, UCDBase identifies physiologically relevant marker genes underpinning the transition from homeostatic renal function to chronic inflammation and fibrosis, while simultaneously capturing the complex interplay between fibroblasts, t cells, and immunosuppressive macrophages.

### Robust malignant subtype identification and cancer feature attribute analysis

Dysregulation of gene expression programs is a hallmark of cancer[36], as such we expected that deconvolution of nonmalignant cells from cancerous cells using transcriptional profiles was possible. We sought

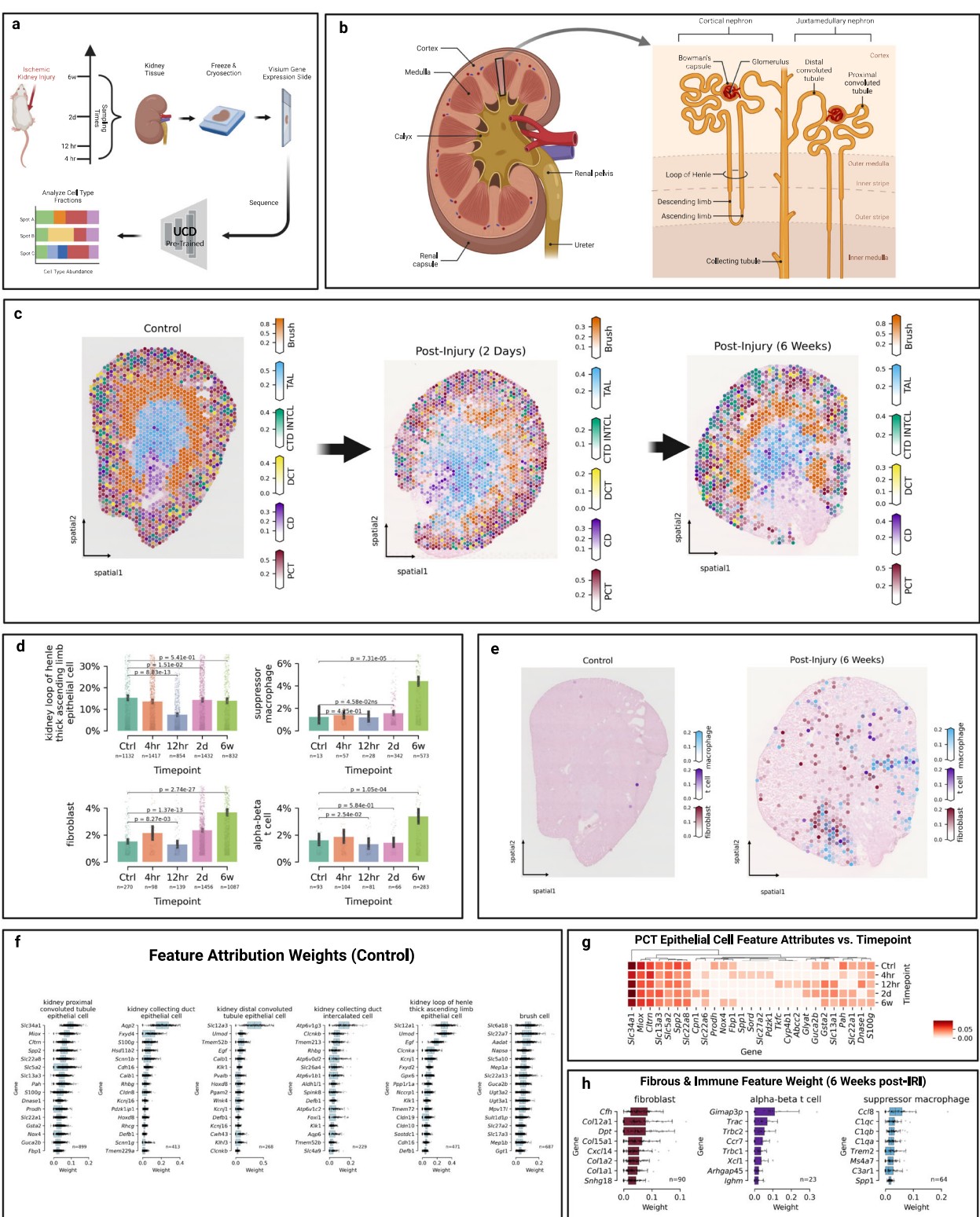

to determine UCDBase's cancer detection and subtype classification performance.

Testing UCDBase's sensitivity to malignant vs. normal tissues, we deconvolved bulk RNA samples from GTEx ($n = 7,851$) and TCGA ($n = 10,459$), predicting samples to be 97.3% vs 74% non-malignant ($p = 0$) when comparing median values of GTEx and TCGA samples, respectively (see Supplementary Fig. 14a right). A notable outlier prediction is seen among GTEx liver samples (see Supplementary Fig. 14a left), which can be attributed to sample-specific pathological,

preprocessing, or quality control factors (or UCDBase training data label misannotation between non-malignant hepatocytes and liver hepatocellular carcinoma (LIHC)). Using deconvolved TCGA data spanning 18 cancer subtypes matched between UCDBase and TCGA, we re-normalized malignant cell results independently of non-malignant cell types to predict cancer subtypes (see Supplementary Fig. 14b). UCDBase achieved a micro-average AUC of 0.889 across all cancers (see Supplementary Fig. 14c), indicating strong classification capability.

**Fig. 2 | UniCell deconvolves mouse kidney undergoing ischemic reperfusion injury. a** Five publically available spatial transcriptomics samples were acquired representing kidney cross sections taken from mice at different stages of ischemic renal reperfusion injury (IRI), and analyzed using UCDBase to determine predicted cell type compositions. A visual summary of the experimental conditions and sample processing is provided. **b** Overview of critical kidney anatomy and general spatial localization of key kidney cell types is shown as a reference. **c** Spatial deconvolution and distribution of select cell types of the *murine* kidney across different time points ($n = 1$ spatial sample at each time point) following IRI. **d** Bar plots of average predicted fractions (*y*-axis) for select cell types deconvolved from spatial transcriptomics samples taken at different time points (*x*-axis) following IRI. Sample sizes are shown beneath each compared condition, representing individual spatial capture spots. Spots with <0.5% reported fraction of a given cell type were excluded from analysis. Bar height denotes the average predicted cell type fraction for each cell type across conditions. Error bars denote 95% confidence interval (CI). *P*-values indicate the significance of difference between groups evaluated using an unpaired two-sided Wilcoxon rank sum test, with Benjamini-Hochberg correction for multiple comparisons (Source Data File−(**d**)). **e** Spatial predictions of fibrotic and immune infiltrate before and after IRI ($n = 1$ spatial sample at each time point). **f** Box plots of feature attribution weights (*x*-axis) for genes (*y*-axis) indicative of select cell types predicted to be present at the control ($n = 1$) timepoint. Sample sizes represent individual spatial capture spots with at least 10% predicted fraction for that given cell type (Source Data File−Fig. 2f). **g** Changes in feature attribution weights for select genes (*x*-axis) indicating proximal convoluted tubule (PCT) epithelial cell fractions shown across different time points (*y*-axis) following IRI (Source Data File−Fig. 2g). **h** Box plots of feature attribution weights (*x*-axis) for genes (*y*-axis) indicative of select cell types predicted to be present at the 6-week post-IRI ($n = 1$) timepoint. Sample sizes represent individual spatial capture spots with at least 10% predicted fraction for that given cell type (Source Data File−Fig. 2h). For scale bars in **c** and **e**, these represent the fraction (range 0–1) a given spatial coordinate is predicted to be composed of a given cell type. For boxplots in **f** and **h**, the center line, box limits and box whiskers correspond to the median, first and third quartiles, and the 1.5x interquartile range, respectively. Individual data points are superimposed over each boxplot.

To gain insight into the gene feature profiles learned by UCDBase, we examined the top-5 gene integrated gradient weights for all 1,143,791 primary cancer cells in our training database averaged by subtype (see Supplementary Fig. 15). Examining the results, we see that UCDBase successfully learns gene expression profiles representing unique transcriptome signatures of subtype-specific malignancies. Demonstratively, prostate cancer adenocarcinoma (PRAD) is identified via *NKX3-1*, a distinct marker of prostatic cancers[37], as well as other genes such as *PCA3*, and *FOLH1*. For melanoma (SKCM), UCDBase associates it with the expression of *MLANA*, the melanoma diagnostic antigen *melanin-A*[38], as well as genes such as *TRYP1* and *MTRNR2L2*. Further inspection of the abovementioned gene features and others (see Supplementary Table 2) demonstrates UCDBase learned subtype-specific gene representations that appear to corroborate their relevance as suggested in prior studies.

We next asked how exactly the feature weights learned by UCDBase distinguish, at a pan-cancer level, malignant vs. non-malignant epithelial cells. We performed differential "relevance" analysis to identify top gene feature weights that tended to be overrepresented and/or underrepresented as predictors among malignant vs. non-malignant epithelial cells across all cancer subtypes. In total, 1,365 genes were identified to be typically positively correlated with malignant cells, while 821 genes were identified to be positively correlated with normal epithelial cells. Each gene set was then subject to GO_BIOLOGICAL_PROCESS_2021 gene set enrichment analysis using Enrichr (see Supplementary Fig. 16). Of significance among malignancy-associated genes, we found "inflammatory responses" to be among the highest upregulated geneset (adj. $p = 5.4E-5$). Numerous genesets pertaining to signaling pathways including *PI3K* (adj. $p = 0.022$), *ERK1/2* cascade (adj. $p = 0.024$), and the *MAPK* (adj. $p = 0.016$) cascades were also identified to be significantly upregulated, in addition to angiogenesis (adj. $p = 0.008$). In contrast, normal epithelial cell gene features appear to overwhelmingly favor cell cycle and regulatory machinery, such as "regulation of G2/M transition of mitotic cell cycle" ($p = 2.57e-12$). Overall, these results appear to suggest that UCDBase may interpret an epithelial cell as cancerous if it exhibits the simultaneous expression of inflammatory and pro-proliferative signaling pathways. Further gene set analysis of UCDBase learned representations may yield additional insights into the fundamental biology of cancer and other disease processes.

## Spatial transcriptomic deconvolution of tumor microenvironment

We next elected to deconvolve a diverse set of publically available solid tumor spatial transcriptomic tissues, including Breast Adenocarcinoma (BRCA), Prostate Adenocarcinoma (PRAD), and Colorectal Adenocarcinoma (COAD). Where available, we compared UCDBase deconvolution results to histological annotations performed by certified *human* pathologists to determine relative accuracy of underlying cell type predictions. Feature attribute analysis was performed for all predicted cell types, with pathophysiologic significance elaborated for each gene in Supplementary Table 2 where appropriate.

## Breast adenocarcinoma spatial deconvolution

UCDBase correctly identified the most likely tumor subtype, BRCA, localized across ductal glands consistent with pathologists annotations (see Fig. 3a). There was strong concordance with pathologist-designated fibrous tissue deposits and fibroblast predictions, attributed to numerous well-established extracellular-matrix (ECM) genes including *COL12A1*, a gene previously implicated in pro-inflammatory stromal desmoplasia and tumor progression in several cancers[39]. Endothelial cells were detected throughout the tumor stroma, and particularly showed strong attribution to apelin receptor (*APLN*), a gene involved in maintaining pro-angiogenic states among endothelial cells, possibly indicating active tumor neovascularization [73].

UCDBase identified multiple immune subtypes, including plasma cells, macrophages, and t cells, localizing to regions of pathologist-annotated immune infiltrate. Tumor-associated macrophages (TAMs) were found at or around areas of comedo-like tumor necrosis[40]. T cells were found to be localizing selectively around a distinct malignant duct located center-left of the tissue section, with attributed genes such as immune checkpoint costimulatory receptor *CD28*, as well as *IFIT3*, *CCL5*, and *PLAAT4* implicating an active anti-tumor immune response. *CD28* is required for an interferon-mediated immune response, coinciding with expression of interferon induced response protein *IFIT3*[41]. The potent lymphocyte attractor ligand *CCL5* is reported to be prospectively upregulated in tumor-infiltrating CD4 + t cells following an initial immune stimulation to maintain t cell infiltration[42]. Furthermore, phospholipase A / acetyltransferase 4 (*PLAAT4*) has been identified as loosely expressed in t cells to support the adaptive immune response[43].

Interestingly, our model strongly implicates *CXCL9* in the prediction of t cells, which is traditionally believed to be secreted by tumor cells themselves or TAMs to drive t cell recruitment[44]. When overlaying gene expression of *CXCL9*, *CD3D* (t cells) and *CD68* (macrophages) (see Supplementary Fig. 17), we see moderate spatial correlation with *CXCL9* and *CD3D* ($r = 0.4$, $p = 3E-29$) along the tumor-stromal interface, and weaker correlation with *CD68* ($r = 0.17$, $p = 1E-10$). We hypothesize that cell-free RNA originating from apoptotic tumor cells in proximity to tumor infiltrating t cells may be captured during single cell encapsulation for sequencing. As the t cell category of UniCell's training data is a generalized category encompassing 191,425 cells of varying possible subtypes and originations, some of which may be tumor-associated, this may be reflected in our results when analyzing cancer datasets. Nevertheless, we see an active image

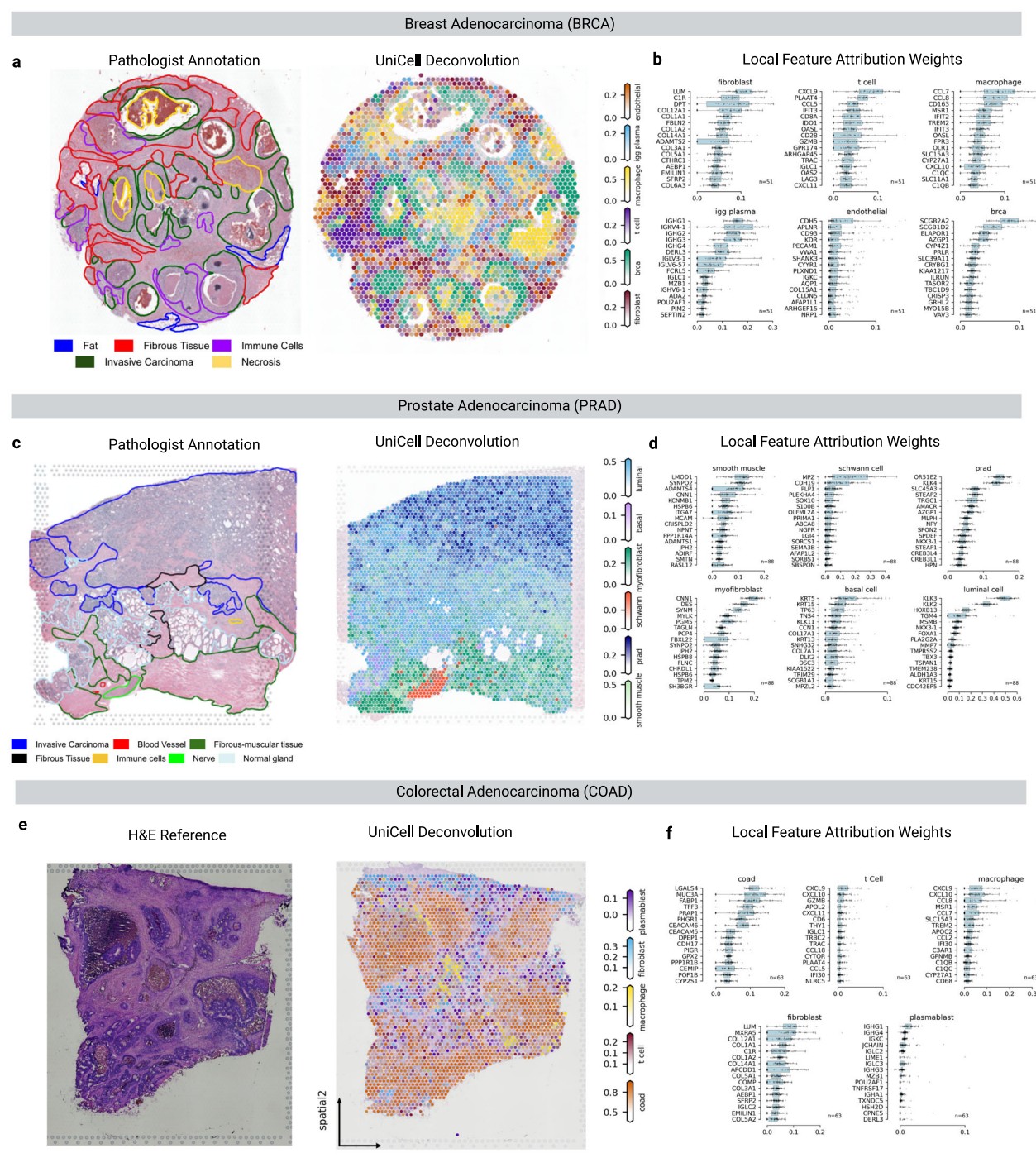

**Fig. 3 | UniCell allows for deconvolution of tumor microenvironments across varying cancer subtypes with unique histologic features. a** (left) Hematoxylin & Eosin (H&E) stained section of a breast invasive adenocarcinoma (BRCA) sample with human-derived pathological annotations (provided with source data) overlaid. (right) UniCell Deconvolve Base (UCDBase) predicted distribution of key cell types in the tumor microenvironment for a sequential section derived from the same sample (n = 1). **b** Box plots of feature attribution weights (x-axis) for genes (y-axis) indicative of select cell types predicted to be present in the BRCA spatial sample. Sample sizes represent the top 2% (n = 51) of individual total spatial capture spots by predicted fraction for that given cell type (Source Data File−Fig. 3b). **c** (left) Hematoxylin & Eosin (H&E) stained section of a prostate adenocarcinoma (PRAD) sample with human-derived pathological annotations (provided with source data) overlaid. (right) UCDBase predicted distribution of key cell types in the tumor microenvironment for a sequential section derived from the same sample (n = 1). **d** Box plots of feature attribution weights (x-axis) for genes (y-axis) indicative of

select cell types predicted to be present in the PRAD spatial sample. Sample sizes represent the top 2% (n = 88) of individual total spatial capture spots by predicted fraction for that given cell type (Source Data File−Fig. 3d). **e** (left) Hematoxylin & Eosin (H&E) stained section of a colorectal adenocarcinoma (COAD) sample. (right) UCDBase predicted distribution of key cell types in the tumor microenvironment for a sequential section derived from the same sample (n = 1). **f** Box plots of feature attribution weights (x-axis) for genes (y-axis) indicative of select cell types predicted to be present in the COAD spatial sample. Sample sizes represent the top 2% (n = 63) of individual total spatial capture spots by predicted fraction for that given cell type (Source Data File−Fig. 3f). For scale bars on the right-side of **a**, **c**, and **e**, these represent the fraction (range 0–1) a given spatial coordinate is predicted to be composed of a given cell type. For boxplots in **b**, **d**, and **f**, the center line, box limits and box whiskers correspond to the median, first and third quartiles, and the 1.5x interquartile range, respectively. Individual data points are superimposed over each boxplot.

of the breast tumor microenvironment rapidly painted by UCDBase, whereby stromal and immune cellular components react to an ever-changing environment driven by active malignancy.

## Prostate adenocarcinoma spatial deconvolution

Turning our attention to prostate cancer (see Fig. 3c), UCDBase robustly distinguishes the tumor subtype, *PRAD* (Prostate Adeno-carcinoma), and localizes malignant cell signatures within the Invasive Carcinoma region denoted in Fig. 3c-left, with nonmalignant luminal epithelial / basal cells in the lower-left region designated as "Normal Gland". Fibromuscular zones outlined in green show distributions of myofibroblasts and smooth muscle cells. This sample contained a nerve fiber cross section, which UCD detected as schwann cells, the myelinating cells of the peripheral nervous system[45]. PRAD is widely considered to be an immunologically "cold" tumor, compared to immunologically "hot" cancers such as melanoma[46,47]. Supporting this, UCDBase did not detect meaningful presence of immune cells in the tested spatial section, and likewise we see PRAD ranking at the lowest end of absolute immune cell fractions among TCGA data deconvolved with UCDBase (see Supplementary Fig. 18).

Changes seen in prostate stromal tissue induced by carcinogenesis are mediated by cancer-activated fibroblasts (CAFs) adopting a myofibroblast-like phenotype[48]. Differentiating between myofibroblasts and conventional smooth muscle cells (SMCs) can be difficult as this phenotype is thought to reflect a continuum spanning conventional fibroblasts to mature prostatic SMCs[49]. Consequently, UCDBase showed overlapping gene attributions used to differentiate these two highly-related cell types (see Fig. 3d).

Feature attributes reveal how UCDBase learned to distinguish normal from cancerous prostate cells. Normal prostatic luminal epithelium was associated with *KLK3* expression (see Fig. 3d). *KLK3* encodes Prostate Serum Antigen (PSA), the most commonly used serum biomarker for prostate cancer despite suffering from low sensitivity due to its universal expression by both normal and malignant prostate cells. UCDBase instead delineates prostate malignancy to *KLK4*, an intracellular kallikrein localizing to the nucleus providing markedly different functions from other KLK family genes[50]. Studies comparing KLK gene expression between prostate cancer and healthy controls have shown stronger statistical correlations between malignancy status and *KLK4* compared with *KLK3*[51].

## Colorectal adenocarcinoma spatial deconvolution

Lastly, we examine UCDBase's deconvolution of colorectal adeno-carcinoma (COAD, see Fig. 3e-right), and we can see clear localization of COAD malignant cells across presumptive tumor nodules shown in the unannotated H&E section in Fig. 3e-left. The stroma surrounding colorectal tumors has been shown to contain uniquely high proportions of infiltrating plasmablasts, a rapidly-dividing intermediate cell state representing activated B cells transitioned into mature, non-dividing plasma cells that function in an immunosuppressive role, which UCDBase readily detects in this sample[52]. Additional immune infiltrates identified by UCDBase include macrophages and t cells sitting among fibroblast cells, highlighting the significant stromal immune responses commonly associated with pro-inflammatory tumor microenvironments.

## Detecting cell type compositional changes in pathological bulk RNA-seq data

Given that scRNA-Seq and spatial transcriptomics remain cost-prohibitive for large-scale translational studies, bulk RNA-Seq data continues to dominate most clinical analyses. We tested UCDBase's bulk-RNA-Seq ability to deconvolve bulk RNA-seq data to reveal pathologic changes in cellular fractions. Feature attributes for each predicted cell type are shown in Supplementary Fig. 19, with detailed analysis of each feature's cellular relevance in Supplementary Table 2.

## Increased fibromuscular tissue deposition in idiopathic pulmonary fibrosis

Idiopathic pulmonary fibrosis (IPF) is a chronic lung disease characterized by the progressive inflammation, damage, and subsequent deposition of fibromuscular tissue into the lung interstitial space, and a corresponding destruction of the alveolar epithelium leading to a reduction in gas-exchange efficacy (see Fig. 4a)[53]. Acute lung injury (ALI), also known as acute respiratory distress syndrome (ARDS), is characterized by transient damage to the gas-exchange apparatus often induced by viral infection, and features significant fibrous tissue deposition as part of the tissue healing process[54].

Comparing Normal, ALI, and IPF tissues (see Fig. 4b), we saw significant reductions in fraction of Type II and Type I pneumocytes (ATII & ATI cells) in chronic IPF patient lungs ($p = 3.33E\text{-}09$ ATII, $p = 5.16E\text{-}08$ ATI), with no difference seen between Normal and ALI ($p = 0.705$ ATII, $p = 0.058$ ATI). This is consistent with the pathophysiologic destruction of alveolar epithelial cells in IPF. Fibroblast fractions were considerably higher for both ALI ($p = 2.77E\text{-}03$) and IPF ($p = 2.41E\text{-}07$) patients compared to normal controls, consistent with the role that excessive fibroblast proliferation plays in IPF pathogenesis[55]. We note a significant increase in smooth muscle cell fractions ($p = 1.55E\text{-}07$), defined by markers such as myosin heavy chain 11 (MYH11), occurring only in IPF patients. Pulmonary hypertension (PH) is a common secondary sequelae to IPF, whereby excessive vascular smooth muscle deposition leads to elevated arterial pressure and potentially fatal cardiopulmonary consequences[56]. Interestingly, we also saw a distinct increase in monocyte fractions for IPF patients ($p = 7.65E\text{-}08$), a finding not seen in ALI. It has been previously reported that elevated monocyte count is associated with IPF progression and may play a role as a useful prognostic biomarker[57].

## Reduction of pancreatic Beta cells in Type II diabetes

Type II diabetes mellitus (T2DM) is a disease characterized by the progressive increase in cellular insulin resistance, leading to a state of persistent hyperglycemia causing a chronic increase of insulin production[58]. The production stresses placed on pancreatic beta cells, responsible for insulin production in the body, eventually lead to apoptosis and selective reduction in beta cell fractions among pancreatic islets (see Fig. 4c)[59].

As T2DM progression exclusively impacts beta cells, we expected to see differences in cell type fractions with respect to disease status only among this cell type. Indeed, we noted a clear, statistically significant decline in pancreatic beta cell fractions ($p = 1.57E\text{-}03$) between normal and diabetes status (see Fig. 4d), with a downward trend ($p = 0.0346$; non-significant after correcting for multiple comparisons) among pre-diabetes patients correlating with disease progression. Beta cell fraction was not correlated to age in this cohort ($p = 0.67$, see Supplementary Fig. 20), although the rate of beta cell proliferation is known to decrease as age increases in the general population[60]. Examining other subpopulations of cell types present in pancreatic tissue (see Fig. 4c), we saw no significant differences in Alpha, Delta, and PP (gamma) cells, and similarly no differences in acinar and ductal cells forming the pancreatic glands.

## Reduced oligodendrocyte fractions in chronic multiple sclerosis

Multiple sclerosis (MS) is a chronic autoimmune disease affecting the central nervous system characterized by chronic inflammation induced by neural lymphocytic infiltration, which leads to progressive destruction of oligodendrocytes, the cells responsible for production of the myelin sheath (see Fig. 4e)[61].

We saw significantly reduced oligodendrocyte fractions ($p = 1.25E\text{-}04$) comparing control and active multiple sclerosis (MS) lesions (see Fig. 4f). No significant changes to cortical neuron or neural progenitor cell fractions were noted; however, a weak trend ($p = 0.041$;

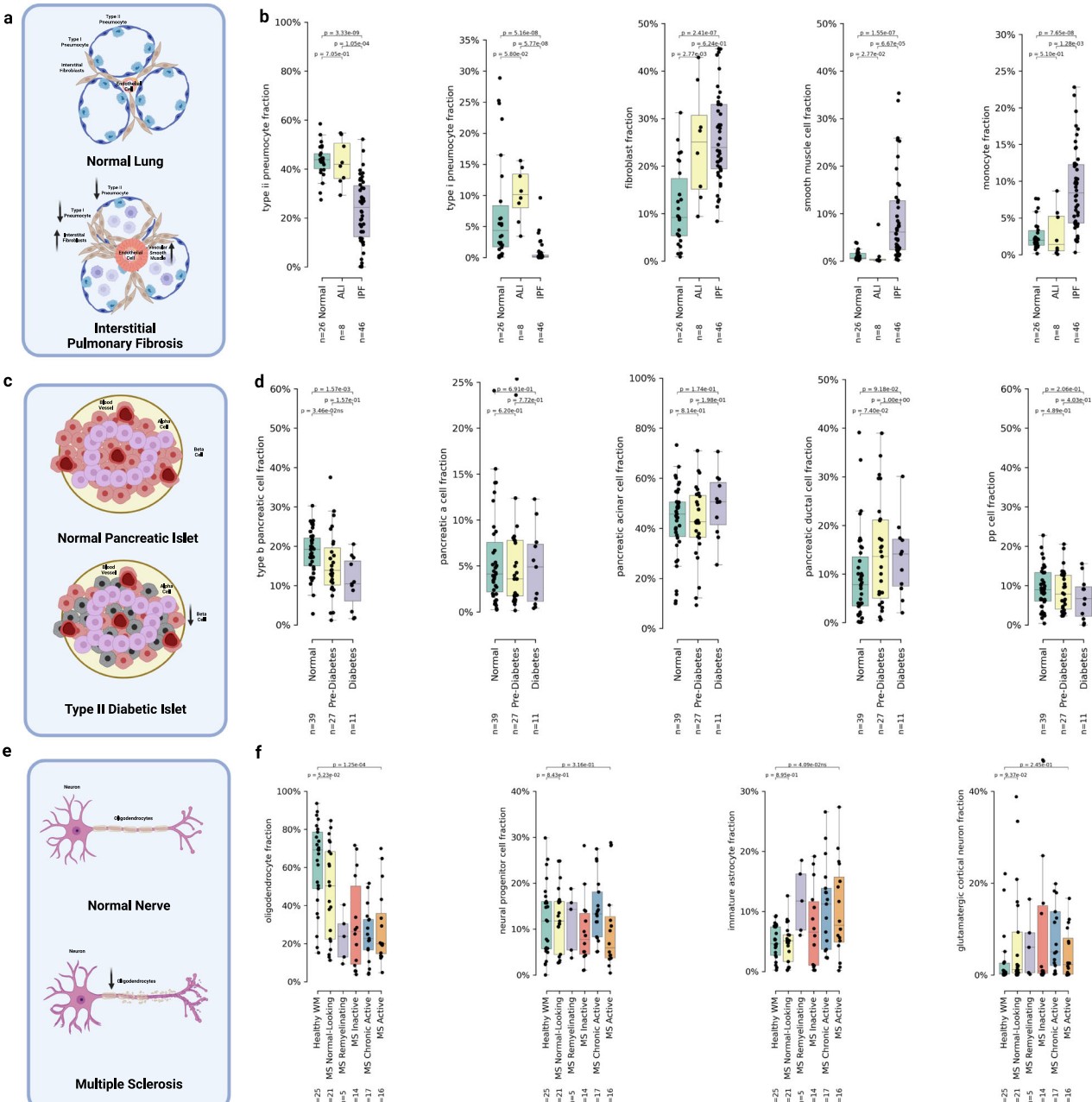

**Fig. 4 | UniCell resolves expected pathophysiological changes in cellular fractions from Bulk RNA-sequencing data. a** Visualization summarization basic pathophysiology of interstitial pulmonary fibrosis and potential shifts in cell type fractions. **b** Box plots of cell type fractions predicted by UniCell Deconvolve Base (UCDBase) for key lung cell types (y-axis) stratified by disease state (x-axis) (Source Data File—Fig. 4b). **c** Visualization summarization basic pathophysiology of type ii diabetes and potential shifts in cell type fractions. **d** Box plots of cell type fractions predicted by UCDBase for key pancreatic cell types (y-axis) stratified by disease state (x-axis) (Source Data File—Fig. 4d). **e** Visualization summarization basic pathophysiology of multiple sclerosis and potential shifts in cell type fractions.

**f** Box plots of cell type fractions predicted by UCDBase for key brain white matter cell types (y-axis) stratified by disease state (x-axis) (Source Data File—Fig. 4f). For all boxplots shown in **b**, **d**, and **f**, the center line, box limits and box whiskers correspond to the median, first and third quartiles, and the 1.5x interquartile range, respectively. Sample sizes for each stratification across all dot plots are shown below x-axis labels, with individual data points being patient samples and superimposed over each boxplot. For all boxplots shown in **b**, **d**, and **f**, P-values indicate the significance of difference between groups evaluated using an unpaired two-sided Wilcoxon rank sum test, with Benjamini-Hochberg correction for multiple comparisons.

non-significant after correcting for multiple comparisons) showing increase in immature astrocytes between control and active MS was found. The proliferation of immature macroglial cells such as astrocytes has been associated with the neurotoxic effects of chronic inflammation induced by multiple sclerosis[62].

Overall, we demonstrated that UCDBase is capable of faithfully recapitulating pathological changes in cell type fractions across a wide range of disease states. This robustness coupled with the validation

offered by feature analysis makes UCDBase a promising tool for the analysis of other pathologic bulk RNA-Seq datasets.

**Rapid cell type annotation and disease subtyping in non-small cell lung cancer scRNA-seq data**

Given strong performance across spatial and bulk RNA-seq tissues, we leveraged UCDBase to assist in basic cell type annotation of a non-small cell lung cancer scRNA-Seq dataset (see Fig. 5a and

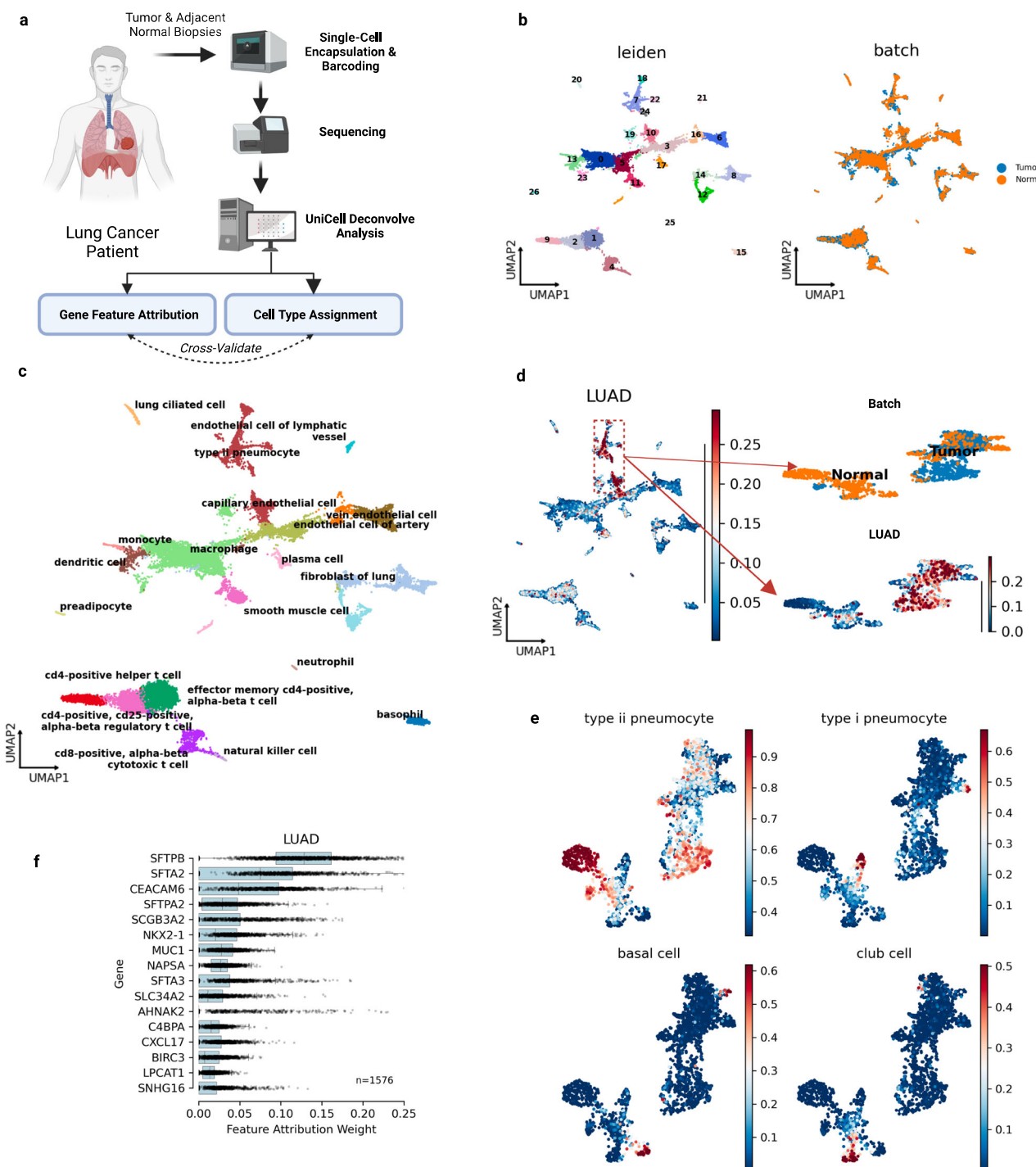

**Fig. 5 | UniCell assists in rapid annotation of an integrated scRNA-seq Non-Small Cell Lung Cancer (NSCLC) dataset. a** Visualization demonstrating the basic steps underlying NSCLC sample collection, processing, and analysis using UniCell Deconvolve Base (UCDBase). **b** UMAP visualization of *human* lung cancer biopsy single cells, annotated by unsupervised leiden cluster (left) and sample of origin (right). **c** UMAP visualization of cell type labels applied for each leiden cluster using UCDBase deconvolution results to guide annotation. **d** UCDBase predictions are used to separate normal from malignant epithelium. UMAP visualization showing probability of malignant lung adenocarcinoma (LUAD) cells initially co-clustering with cells labeled as normal epithelium (left). Re-clustering select subpopulation reveals two major clusters separating by sample of origin, Adjacent Normal or Tumor (right upper). Visualizing UCDBase LUAD probabilities on re-clustered cells demonstrates Tumor-specific cluster contains the majority of predicted LUAD malignant cells. **e** UMAP visualization showing probabilities of four major lung normal epithelial cell types distributed across re-clustered cells. **f** Box plots of feature attribution weights (*x*-axis) for genes (*y*-axis) indicative of LUAD malignant cells learned by UCDBase. Sample size ($n = 1576$) reflects the total number of single cells annotated as malignant LUAD. For boxplots, center line, box limits and box whiskers correspond to the median, first and third quartiles, and the 1.5x inter-quartile range, respectively. Individual data points representing single cells are superimposed over each boxplot (Source Data File–Fig. 5f).

Supplementary Fig. 21a), validating assigned cell types using feature attribution analysis (see Supplementary Fig. 23) followed by a literature analysis of identified markers (see Supplementary Table 2).

Examining the annotated clusters (see Fig. 5b, c) further, we sought to identify malignant LUAD cell subpopulations among predicted epithelial cells, which were found by UCDBase to likely be located within leiden clusters 18, 7, 22, 24, and 10 (see Fig. 5d-left). Because these clusters appeared intermixed with normal epithelial cells, we reclustered this subset of cells at higher resolution to reveal separations between malignant and nonmalignant cells (see Fig. 5d-right). We saw clear separation of cell clusters by biopsy status, indicating most likely that tumor tissue contained a predominance of malignant cells. Indeed, UCDBase predicted a higher probability of lung adenocarcinoma (LUAD) cells across the tumor biopsy derived cell clusters, with little to no malignant signal across cells derived from adjacent normal. To orthogonally validate malignancy predictions, we performed copy-number variation (CNV) inference, using a combination of smooth muscle, fibroblast, lung ciliated, and endothelial cells as reference controls, finding that UCDBase malignancy predictions overlapped estimated increased copy number variation (see Supplementary Fig. 21b, c). We quantified this relationship, finding considerably positive and significant correlation (spearman $r = 0.39$, $p = 1.7\text{E-88}$) between malignancy probability and average CNV score per cell (see Supplementary Fig. 21d).

Some LUAD feature attributes (see Fig. 5f) were found to mirror surfactant genes related to type II pneumocytes, unsurprising as ATII cells are believed to be the cell of origin of LUAD[63]. A major malignancy-specific feature identified was carcinoembryonic antigen 6 (*CEACAM6*), known oncogenic gene overexpressed in numerous cancers including non-small cell lung (NSCLC), colon, and breast cancers[64]. Additional NSCLC-related genes identified include *NKX2-1*, a key transcription factor involved in early lung development and diagnostic marker for LUAD[65]. Non-malignant epithelial cells (see Fig. 5e) were clearly assigned to lung-related cell types with straightforward feature attributes (see Supplementary Figu. 22) corresponding to established cell type markers (see Supplementary Table 2 for details). Overall, UCDBase enabled the rapid and accurate annotation of a complex NSCLC patient case, with feature attribute analysis allowing for prospective validation of cell type assignment, in addition to delivering contextual information pertaining to the biological processes underpinning the data itself.

## Discussion

In this work, we presented UniCell: Deconvolve Base, a universal, context-free cell type deconvolution tool for transcriptomic data that integrates the entirety of publicly-available scRNA-Seq data into a single unified training dataset for deep learning applications. Our corpus of 28 M fully-annotated single cells enables UCDBase to generate accurate cell type fraction predictions without the need for tissue or disease-context, enhancing its ability to explore and discover biological phenomena across all major subtypes of transcriptomic data. UCDSelect on the other hand, allows for context-specific deconvolution using user-defined cell signatures by leveraging transfer learning of UCDBase features.

We demonstrate that UCDBase and UCDSelect are capable of producing highly accurate deconvolution predictions using both synthetic scRNA-Seq mixtures and real-world spatial transcriptomics data, that are comparable and/or superior to state-of-the-art methods.

We highlight UCDBase's deconvolution of the dynamics underpinning ischemic renal injury, in addition to the tumor microenvironment from differing cancer subtypes. We show how UCDBase can be leveraged together with feature attribute analysis to uncover pathophysiologic responses in bulk RNA-Seq datasets for: idiopathic pulmonary fibrosis, type II diabetes mellitus, and multiple sclerosis. Lastly,

we leverage UCDBase to assist in cell type annotation of a scRNA-Seq NSCLC dataset.

We acknowledge that cell type labels provided by authors either directly via metadata or indirectly in studies may not be entirely accurate, and/or lack specificity with respect to labeling of distinct cell subtypes (i.e. labeling an immune cell as a *CD4 + t cells* vs. *CD4 +* effector memory t cell). Prediction specificity can be improved markedly by increasing the granularity of cell type label assignments via enhanced data integration to refine our primary training data corpus.

We demonstrate in Supplementary Fig. 10 how high levels of gene dropout (80%+), and by extension absolute sequencing depth, can negatively affect model performance. This limits the applicability of UCDBase and UDCSelect towards targeted / in-situ spatial transcriptomics platforms. Improvements in training data augmentation by direct modeling of count downsampling in real time during training will enable future iterations of UCDBase to be more robust to dropout, and capable of better handling in-situ data.

UCDBase was designed using empirical evaluation of a range of hyperparameters concerning layer sizes, depth, and regularization parameters. Future iterations of UCDBase will leverage neural architecture search to iteratively test model layouts across a range of architecture choices and corresponding hyperparameters[66].

The analysis of transcriptomic data is a challenging process necessitating significant time investment by end users to generate biologically plausible conclusions. Cell type annotation and deconvolution in the cases of scRNA and ST, respectively, are often laborious processes. With UniCell: Deconvolve Base (UCDBase), we provide a one step solution to this problem, generating accurate predictions across three transcriptomic data modalities, scRNA-Seq, bulk RNA, and ST without the need for additional user input. With UniCell: Deconvolve Select (UCDSelect), we enable deeper exploration of data, leveraging the benefits of transfer learning from a global pre-trained model, together with the contextual specificity of user-defined cell signatures.

We believe that the UCD tools suite, as a consequence of its comprehensive nature, ease of use, and speed, will accelerate the ability for the broader research community to conduct complex science, understand the cellular context underpinning diseases, and drive the development of therapeutics to address them.

## Methods

### Ethics and oversight statement

The analysis in this manuscript was conducted using predominantly publicly available datasets. Prospective tissue samples acquired for single cell analysis were collected through the Mount Sinai Hospital (MSH) via the Mount Sinai Pathology Core Facility. Approval for this study was granted by the Mount Sinai Lung Tissue Utilization Committee. The tissue studied was acquired under the Institutional Biorepository protocol (12-00145) which allows for collection of excess surgical tissue that is not needed for diagnostic purposes, to be used for research. For this protocol, informed consent is based on specific language included in the general surgical consent, which all patients sign prior to surgery. All tissue is distributed in a de-identified fashion.

### UCDBase model overview

UCDBase is a Deep Neural Network (DNN) with 281,397,066 trainable parameters that accepts normalized RNA expression input and outputs predicted cell type fractions (see Fig. 6b). Below we describe the UCDBase architecture in detail and provide a rationale for key design choices.

### Primary data input and preprocessing

UCDBase accepts, by design, nearly all coding and non-coding human genes, for a total input size of 28,867 genes. Our approach takes advantage of the fact that DNNs, by nature of their

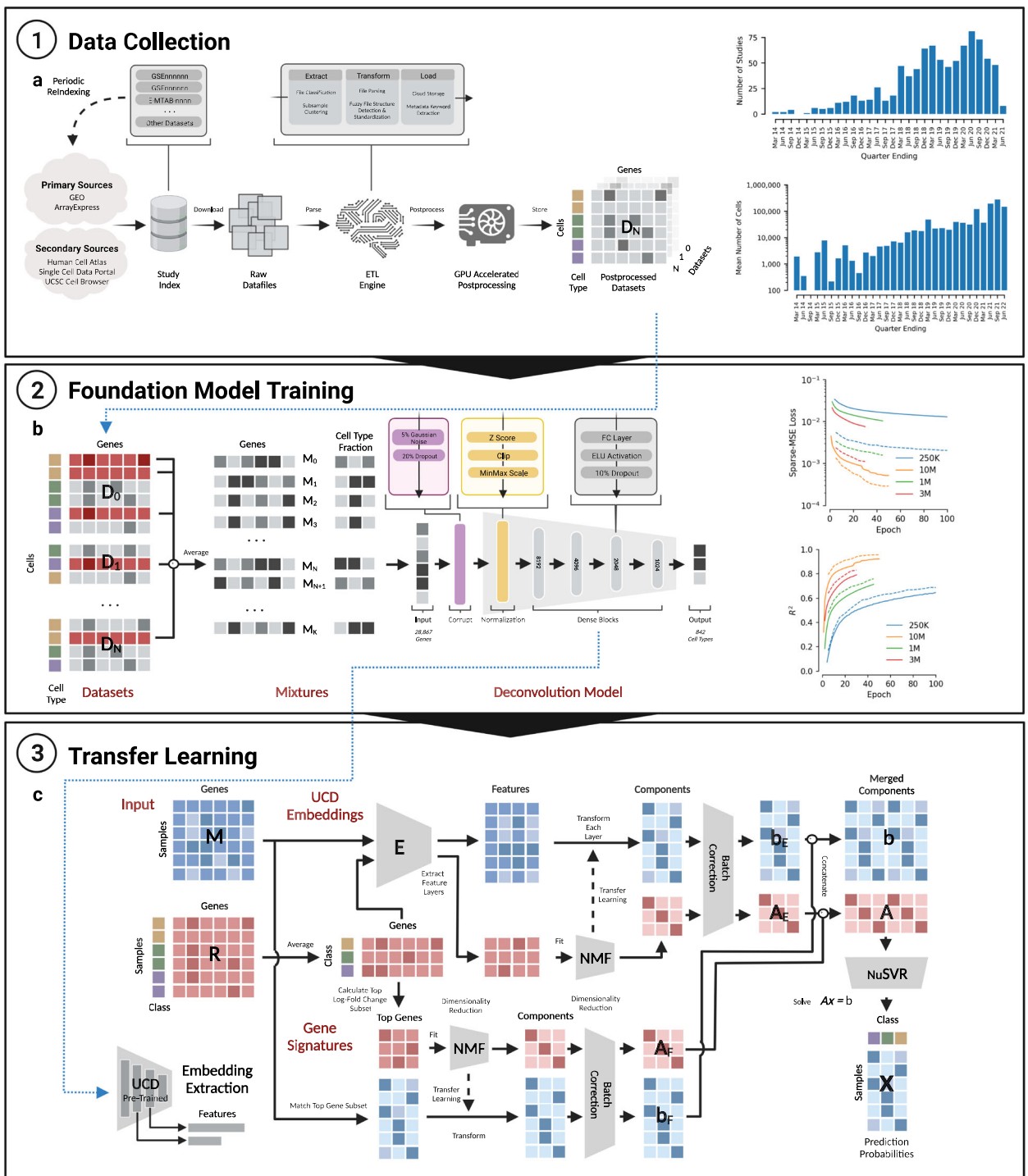

**Fig. 6 | Summary of UniCell data collection, training mixture generation, foundation model architecture, and transfer learning strategy. a** Depicted on the left is a flow chart summarizing the training data collection strategy. Candidate studies are first indexed from several primary and secondary data sources. Raw data is downloaded from respective source locations, and processed through an ETL engine where the output represents a standardized single cell count matrix. GPU accelerated post processing is performed, resulting in a normalized single cell expression profile. The number of studies indexed and total number of cells profiled (*y*-axis) is shown as a histogram on the right, within 3 month interval buckets (*x*-axis). **b** Each normalized single cell expression profile is utilized to form training data in the form of single cell mixtures, whereby random subsets of cells from across studies are selected (see flow chart on left) and averaged together to create mixed expression vectors of known cell type fractions. Expression vectors are fed into a deep learning model trained to predict the known cell type fraction.

The basic elements and structure of the UniCell Deconvolve Base model are shown in the flow chart. On the right, an overview of the training process is shown. The *y*-axis represents either model loss or coefficient of determination ($R^2$) while the *x*-axis represents training epoch, where one epoch represents a single full cycle through training dataset. Each colored line corresponds to a different size of training dataset (250 K, 1 M, 3 M, or 10 M synthetic mixtures). Solid lines represent model performance on the training dataset, while dashed lines represent model performance on test dataset. **c** Users have the option of supplying a contextualized reference profile, which is used in conjunction with embeddings obtained from UCD Base acting as a universal cell state feature extractor. A regression model is then trained using processed embeddings, yielding a fine-tuned transfer learning model applicable to user-specific use cases. Details of the transfer learning model architecture are shown in the corresponding flow chart.

overparameterization, do not suffer reduced performance from multicollinearity[67]; a phenomenon exhibited when one or more model input values (e.g. gene expression) are highly correlated that can negatively impact machine learning model performance. We hypothesized that an overparameterized input space would buffer performance against sparsity due to tissue heterogeneity and/or technical resolution exhibited by current transcriptomics platforms, allowing UCDBase to rely on alternate, non-canonical genes for cell type prediction in cases where canonical markers are not captured insufficiently sequenced.

Inputs, consisting of a 1-dimensional vector of gene expression values representing a single-cell or mixture of cells (hereby referred to as input), are normalized on a per-input basis. We surmised that UCDBase would be able to infer cell type signatures using relative differences in expression signals, and that per-input normalization would make the model more robust to differences in feature scales between training and test data. Gene expression counts are first normalized to 10,000, followed by log2 scaling so as to reduce the effect of heteroscedasticity on expression distribution. Each sample is z-scored to standardize variance across features. Lastly, we apply min-max scaling to rescale each feature value from 0 to 1, which is then used as input into UCD.

To further reduce reliance on canonical markers and limit the impact of sparsity, we introduced a two-step corruption process to our normalized sample inputs during model training. We first inject 5% gaussian noise to the normalized expression profile of each gene (e.g. A normalized expression value of 0.8 from a given gene $i$ will range anywhere from 0.75–0.85 following 5% gaussian noise injection). This is followed by a dropout layer, where 20% of input values are randomly set to zero. We reasoned that a combination of noise and dropout would further encourage the DNN to learn more complex representations of cell types that are robust to noise and missing genes.

### Intermediate layers

The core of UCDBase consists of four fully connected dense layers of 8192, 4096, 2048, and 1024 neurons using an exponential linear unit (ELU) activation function. Baseline characteristics of the model architecture, layer sizes, and overall depth of the network were determined through sequential, empirical evaluation of preliminary models of varying size on subsets of the final training dataset. In brief, we randomly subset the core training dataset to between 250 K to 1 M mixtures, and repeatedly trained models out to 10 epochs to determine the effect of sequential changes to individual parameters on deconvolution performance. We noted that model performance in the first few epochs (see Fig. 6b-right) was indicative of eventual convergence accuracy, making this a suitable proxy for rapid iterative optimization.

### Output and post processing

The final layer of the model is a dense layer of 840 neurons, corresponding to all cell types available in our training database to-date, with a softmax activation function yielding cell type fraction estimates summing to 1. No additional regularization is applied to the output layer, for it was found to reduce overall performance.

The cell types in the resulting deconvolution sit at varying levels of cellular specificity hierarchies (i.e. 't cell' vs. 'cd4-positive, alpha-beta t cell'), a consequence of leveraging author-derived annotations and/or low-confidence in more specific labels. In order to account for prediction biases induced by this uncertainty (i.e. some t cells may in fact be cd4 + t cells, while all cd4 + t cells are themselves t cells), we employ a belief propagation (BP) step during output post processing. BP involves projecting initial cell type fraction estimates onto a cell type hierarchy subset from the Cell Ontology (see Supplementary Fig. 4)[68], and summing probabilities upwards along the directed tree structure. In such a way, fractional probabilities assigned to certain cell type subclasses are captured to yield higher confidence estimates of deconvolution fractions for more generic cell types.

### Generation of training data

UCDBase is trained using mixtures of simulated RNA-seq data (pseudobulk mixtures) generated from scRNA-Seq data. The process of generating a mixture is described in the following steps: (1) **The total number of cells (T) comprising a mixture is selected**. Given our desire to develop a model robust to both low-input (i.e. single cell / ST) and high-input (i.e. bulk RNA) samples for deconvolution, we randomly selected a value from 1 to 10,000 with uniform probability. (2) **The number of unique cell types (N) in a mixture is chosen**. We selected anywhere from 1 to 32 cell types to appear in a given mixture with uniform probability. The maximum value of 32 cell types (although parameterizable for future training) was assigned after analyzing the cellular diversity of all curated scRNA-Seq datasets, and taking the nearest log2 value of the 95% percentile for the number of unique cell types per dataset. Selecting cell types with uniform probability has the effect of oversampling cells with low representation in the dataset, which improves model performance on rare classes. (3) **The mixture fraction ratios F for N cell types are assigned**. We assigned a random fraction ratio $F_i$ for each cell $N_i$ in a given mixture, such that all fraction ratios summed to 1. (4) **Expression data for cell types are accumulated and averaged together**. For each cell type $N_i$ in a sample, we randomly selected $N_i*T$ cells of that type from our uniformly preprocessed, integrated scRNA-Seq database. In cases where the required number of cells exceeds the total number of cells of a given type available in the dataset, the maximum number possible were added to the mixture, without duplication. Once all required cells were randomly selected, expression profiles were averaged together with a simple mean, resulting in a pseudobulk RNA expression profile with a known cell type fraction.

### Mixture formation via rapid data integration

The process of pseudobulk sample generation was implemented in python and optimized for high-performance execution using the python numba package (Numba: A high performance python compiler. https://numba.pydata.org/). All hyperparameters T, N, and F were precomputed as described above prior to generating mixtures, and cell type array row locations were pre-indexed to avoid repeat searches and improve performance. A total of 10 million pseudobulk mixtures were generated over the course of 18 h at a rate of 150 mixtures per second using a total corpus of 28 million annotated single cells into a 28,000,000 × 28867 compressed-sparse-row (CSR) matrix, running on a Google Cloud Engine (GCE) n2d-standard-224 virtual machine (VM) instance with 224 vCPU cores and 896GB system RAM. The choice of 10 million pseudobulk mixtures was made by training multiple iterations of UCD with stepwise increases in training dataset size, noting the impact the amount of mixture examples had on model performance (see Fig. 6b-right). We observed an increasing logarithmic relationship between training data size and performance, and determined 10 M mixtures to be the optimal size for initial model evaluation as a tradeoff between model accuracy and training time. Increases in size offered diminishing projected returns with respect to theoretical peak performance (see Supplementary Fig. 1). Ultimately, these training parameters can all be customized as future training sets become more expansive beyond 840 cell types and/or if necessary for extended accuracy in use cases where runtime beyond 18 h is not limiting given the rapid nature of the overall end-to-end training time.

### Single cell dataset curation

The collection and integration of a large annotated scRNA-Seq database is essential to the performance of UCDBase. In this section, we describe the major stages of our data curation process (summarized in Fig. 6a) and highlight technical approaches used to overcome

challenges inherent to operating with integrated high-dimensional data at scale.

## Study indexing

We generated an index of all publicly available scRNA-Seq datasets, leveraging both primary sources such as NCBI Gene Expression Omnibus (GEO) (geo. Home−GEO−NCBI). https://www.ncbi.nlm.nih.gov/geo/ and EMBL ArrayExpress (AE) (EMBL-EBI. ArrayExpress. https://www.ebi.ac.uk/arrayexpress/), as well as numerous secondary source including the UCSC Cell Browser (UCSC Cell Browser. https://cells.ucsc.edu/?), EMBL-EBI Single Cell Expression Atlas (EBI Gene Expression Team− https://www.ebi.ac.uk/about/people/irene-papatheodorou, Single Cell Expression Atlas. https://www.ebi.ac.uk/gxa/sc/home), TISCH[69], and the CZI Human Cell Atlas (Home, https://www.humancellatlas.org/). For primary data repositories GEO/AE, we performed an API-based programmatic keyword search for "scRNA-Seq OR single cell OR single-cell sequencing OR scRNA" to collect an exhaustive list of studies potentially containing scRNA-Seq data. Primary and secondary sources were then manually cross-referenced to eliminate duplicate entries.

At present, our base index contains 2695 studies published between January 2015 and June 2021, as such any studies published before or after this period are not currently included in UCDBase, but will be ingested, indexed and included in training sets for future builds. Examining global trends in publications (see Fig. 6a-top right), we note a steady increase in the number of scRNA-Seq biomonthly binned publications between 2014 and 2021 where data is available. Importantly, we see the number of single cells profiled in experiments increasing from a general average of 100 cells beginning around 2015 to over 10,000 cells per study in 2021 (see Fig. 6a-bottom right). As these trends are only expected to continue increasing, we anticipate a plethora of additional transcriptomic information will become available, the integration of which into global, accessible datasets will further aid in the development of not only machine learning algorithms, but fruitful data reanalysis revealing unique biological insights. We anticipate performing additional study indexing on at least a monthly basis at minimum to allow for integration of recently published studies into model training cycles, but it should be noted that ad hoc retraining can be conducted anytime using public or non-public datasets in <24 h using existing computing infrastructures.

## Data extraction

Each indexed study is passed through an automated data loader customized to each unique input source (i.e. GEO vs. AE) in an attempt to automatically extract scRNA-Seq count matrices. We first categorize all supplementary files associated with a particular study, looking for delimited file type extensions used for either transcriptional data or metadata (e.g.csv,.tsv,.h5,.h5ad,.mtx, etc…). In cases where expression data is stored as multiple files (i.e. 10X Genomics matrix.mtx/barcodes.tsv/genes.tsv triplet format), we successfully match pairs of common filenames by stem using text-similarity unsupervised clustering. Metadata when present, including cell type annotations, is typically found in separated delimited files and is identified by matching filename substrings "meta OR metadata OR annot OR annotation". Files identified as potential expression or metadata are then batch downloaded using the aria2 utility for further processing.

## Data transformation

For each datafile in a study, we attempt to load, parse, and standardize gene expression data, and then match it with any associated metadata (see Fig. 6a). In most cases, expression data is stored in a delimited file structure (i.e.txt,.csv,.tsv formats) where each row and column correspond to cells and genes, respectively, or vice-versa. The major steps in file loading and standardization are: (1) **File delimiters are first identified** based on the most common present in the first line of the file (i.e. tab, space, comma, etc). (2) **We estimate file dimensions** using

a heuristic function that calculates the bytesize of the first N-lines of a given file and compares it to the total file size. (3) Delimited files exceeding 100,000 projected rows or columns are **read using a bespoke lightweight data parser**, SRead, which distributes line reads across a unified thread pool for rapid data loading. Smaller files are read using the python pandas read_table function using the identified delimiter. The final output is yielded as a pandas DataFrame object. (4) **Gene names are standardized to gene symbols** using a comprehensive dictionary of gene IDs, synonyms, and symbols, where we further identify whether or not a row or column in the loaded DataFrame contains gene information and set this as the index or header, respectively. Depending on the initial data frame orientation, we correct orientation to follow tidy data conventions such that rows correspond to cells (observations) and columns correspond to genes (variables). Columns containing string-like characters are assumed to correspond to cell index names or associated cell-metadata, while those containing floating point or integer values are assumed to be expression data. (5) **We attempt to match rows or columns of metadata with standardized row indexes of a given sample file**. If a high degree of concordance is found between a data matrix and files flagged as potential metadata, we assume the file corresponds to cell-level metadata and align both dataframes together for final integration. (6) **Lastly, we convert expression data into compressed-sparse row (CSR) matrices and map them together with align metadata (if-any) using the annotated dataset (i.e. h5ad) library**. These H5-like objects are then uploaded to a Google Cloud Storage (GCS) bucket as unprocessed, standardized data sets suitable for downstream processing.

## Data preprocessing

Before a scRNA-Seq dataset can be utilized, it must undergo additional preprocessing. The most commonly used packages for scRNA-Seq processing and analysis, scanpy (python) and seurat (R), were not originally designed for high-throughput batch processing of thousands of scRNA-Seq datasets. Many computational steps, including covariate regression, batch correction, nearest neighbor calculation, and dimensionality reduction, can take significant time for datasets exceeding 100,000 cells. To enable UCD, we developed scanpyRAPIDS, a single cell analysis framework that enables complete end-to-end GPU-accelerated scRNA-Seq preprocessing. Leveraging the CuML, CuGraph, and CuPy python libraries from RAPIDS.AI (API docs. RAPIDS Docs https://docs.rapids.ai/api), we reimplement the entire standard scRNA-Seq preprocessing pipeline from basic QC through batch correction, dimensionality reduction and clustering residing entirely in GPU memory. Relative performance gains compared to traditional CPU-bound analysis is dependent on both the size of the input data and functional requirements of data preprocessing. For example, our scanpyRAPIDS implementation of the popular harmony batch correction algorithm[70] successfully integrates 209,264 cells from 107 individual samples representing a time course of iPSC induction in 201.1 s on an NVIDIA Tesla T4 GPU, compared to 1204.6 s on a 16-core vCPU instance with 100+GB RAM. This presents a 6-fold speedup in runtime that continues to scale linearly with dataset size.

Using scanpyRAPIDS, all raw H5AD objects from the previous stage are concurrently preprocessed, parallelized across 4 NVIDIA Tesla T4 16GB GPUs. In brief, cells with <200 counts or genes expressed in <3 cells in a dataset were filtered out. Cells with >20% mitochondrial read fraction were assumed to be dead or damaged cells, and filtered out. Cells whose total counts exceeded two times the standard deviation of log-normal total counts for all cells in the sample were assumed to be damaged outliers, and filtered out. Total counts were normalized to 10,000 reads per cell and subsequently log-normalized. Depth normalized log counts are subsequently retained as input vectors for training mixture generation.

Dimensionality reduction and sample-level visualization was performed to better facilitate manual quality control (QC) checks of dataset quality and cell type annotations. As such, highly variable genes (HVG) were calculated, keeping genes with a log-normed mean between 0.0125 and 4, and a minimum dispersion of 0.25. HVGs were then z-score scaled to +/− 10. We regressed the effect of cell read depth (total counts) on expression of each HVG using an CUDA-accelerated ElasticNet regressor. Lastly, we performed PCA, with the number of components determined based on the number of post-filtered cells present in the sample. Nearest neighbor calculation was performed with n_neighbors set to 30. We ran both 2D and 3D UMAP dimensionality reductions, with min_dist set to 0.3. Lastly, unsupervised leiden clustering was performed, with resolution determined, like PCA components, on the number of post-filtered cells in the sample. Post-processed H5AD AnnData objects were then uploaded to a GCS bucket. For cases where multiple samples were preprocessed for a given study, we performed batch correction and re-clustering using the described approach with our GPU-accelerated implementation of the Harmony algorithm[71].

## Data storage
When determining the optimal active data storage format, we had two concerns to address. Firstly, as our total preprocessed data repository contains nearly 1TB of data, which is expected to grow overtime and will need to be shared between team members, local *on-disk* storage would not be practical. Second, the need to rapidly load, inspect, and validate preprocessed data prior to final integration made traditional disk-mapped data formats such as HDF5 (and by extension, H5AD) limiting due to I/O throughput and cloud-access flexibility perspectives. As a result, we designed a bespoke data storage model, Single-CellData (SCD), built on top of the TileDB API. TileDB is a cloud-native data storage solution that integrates with cloud storage solutions such as GCS and S3, with explicit support for multidimensional, sparse array storage and parallel, chunked I/O operations[72]. The conversion of our preprocessed H5AD objects into SCD format allowed us to rapidly access and validate preprocessing quality for our datasets.

## Cell type annotation and label transfer
A total of 1,712 unique studies with 10,000+ associated data files were successfully preprocessed and stored using the above methods. Approximately 20% of preprocessed data had cell type annotations available. Datasets without cell type labels were annotated using a semi-automated procedure involving manual curation of annotations using canonical & publication-derived marker genes, supported by an initial coarse cell type label transfer.

## Annotation of cell types
We sought to first project both annotated and unannotated cells into a common latent space using a deep autoencoder model in order to cluster similar cell types together and transfer labels of nearby known cell types onto unannotated cells. Although manual verification of the annotations was conducted, the use of an autoencoder joint-embedding model significantly accelerated the rate at which annotation could be accomplished.

To that end, we trained a spherical variational autoencoder (sVAE) with 30 latent dimensions on all preprocessed gene expression profiles (see Supplementary Fig. 5). In brief, sVAEs differ from conventional variational autoencoders (VAE) in the use of a non-normal prior distribution for parameter regularization. Early work applying sVAEs to scRNA-Seq data has shown benefits compared to traditional VAE in terms of embedding stability, leveraging the von-Mises-Fisher (vMF) spherical distribution[73]. For our implementation we utilized the PowerSpherical distribution, a related distribution that offers improved numerical stability during model training[74]. Nearest neighbors were determined using cosine similarity relative to 30 embedded latent

dimensions using CuML, followed by unsupervised leiden clustering with the resolution hyperparameter set to 2.

For each of the 4200 initially identified clusters, known cell type classifications were averaged, assigning the most common annotation within a cluster to any cells unknown labels. Preliminary validation of each cluster annotation was performed by decoding the mean embedding vector to obtain a denoised, average gene expression profile for that cluster, and examining the highest expressed genes for correlations between canonical marker genes and predicted class types[75,76]. The process was then repeated, re-grouping cells into high-level subtypes (i.e. b cells, t cells, neuronal cells, etc) to obtain more refined subtype classifications.

For each dataset, cell type assignments were manually compared to available figures published in corresponding studies. Cases where first-pass, automated coarse annotations were too broad, incorrect, or did not match tissue-specific labels found in the study, were each manually identified and corrected. At the time of writing, 898/1,712 studies were verified to pass QC, with an initial focus on the largest and most diverse datasets available. Expression profiles of all cells associated with studies that passed the QC criteria were then averaged across common cell types, and the top 50 differentially expressed genes for each cell type cluster were computed and made available in Supplementary Table 3. Visual inspection of top marker genes cross-referenced with known canonical markers provided empirical evidence of accurate cell type label assignments. In total, just over 28,000,000 single cells are contained in our dataset reflecting 840 unique cell types, including 55 cancer subtypes and 156 distinct cell lines (see Supplementary Fig. 3).

## Training strategy
The UCDBase model described previously was implemented and trained using Tensorflow 2.5.0. We utilized the Adam optimizer for supervised backpropagation with a learning rate of 0.0001 and an effective batch size of 256. Loss was computed using a variation of mean-squared error, sparse MSE. Given that for most examples, the vast majority of true cell type proportions are zero, we found during initial testing that conventional MSE would be artificially inflated, introducing a negative bias on model training. As a result, we opted to mask cell types whose mixture proportions are zero from calculation of MSE for that given sample, in effect focusing the loss function to contextualize towards each training sample.

Pseudobulk training data generated as described previously was serialized into TFRecord objects and saved into a separate GCS bucket, subsequently fed into the UCDBase model using the tf.data API. We trained UCD across 50 epochs over the course of 7 h running a pre-emptible Google Cloud Engine (GCE) a2-megagpu-8g instance, comprising eight NVIDIA A100 40GB GPUs, 96 CPUs, and 680GB system RAM. A train-test-split ratio of 80/20 was selected for training validation, and test validation was conducted every five epochs and subsequently interpolated for visualization. Early stopping was utilized with a patience interval of 4 epochs and a loss delta threshold of 1.25E-5. Learning rate was dynamically lowered by 50% if training loss did not improve by 1E-4 within 5 epochs. Results of model training, as measured by sparse MSE and pearson correlation, are highlighted in Fig. 6b-right. We observed model convergence by epoch 50, at which point the conditions of early stopping were met. Additional analysis of model convergence (see Supplementary Fig. 6) demonstrated that further training using the current training dataset would not yield tangibly relevant increases in model performance.

## UCDSelect model overview
UCDBase was designed to support the unbiased deconvolution analyses of datasets in instances where a reference signature is unavailable or unclear. However, when available, we recognize that following an initial assessment of the cellular composition within a dataset, it would

be beneficial to enable a mechanism for model fine-tuning using a user-specified cell type signature. The contents of the user defined signature could also be determined and supported by the context-free analysis generated by UCDBase. Ideally, this mechanism would work seamlessly with the existing UCBase pipeline, and leverage the pre-trained base model to increase performance with minimal computational overhead. To that end, we propose a transfer learning extension, called UCDSelect (see Fig. 6c), that enables users to leverage the benefits of a pre-trained foundation model together with the specificity of a user-defined cell type signature.

As input, UCDSelect takes the same expression data representing a cell type mixture as UCDBase, with an added reference expression dataset including corresponding cell type labels. Reference data is then averaged across cell types to generate a mean expression signature for user-specified cell states. The input data from both mixture and reference are then fed into our pre-trained UCDBase model, with outputs consisting of the middle two dense layers of the neural network, of dimensions 4096 and 2048, respectively. The two feature vectors corresponding to the reference signature are then independently subject to dimensionality reduction using non-negative matrix factorization, an approach similar to that employed by the SPOTlight deconvolution algorithm, albeit with the input representing model features rather than raw gene expression values[18]. Each NMF model is fit using the reference, and used in turn to then transform the mixture. We utilize the Combat algorithm[77] to perform batch alignment on the resulting NMF components so as to improve distribution concordance between reference and mixture, which has been used successfully in other reference-driven techniques, such as CIBERSORTx[12]. We repeat the above decomposition process using feature-selected gene expression values, and generate a final merged set of batch-corrected NMF components. We found that in most cases UCDBase features achieve improved relative deconvolution performance, the integration of both extraction techniques leads to slightly higher overall accuracy with negligible performance degradation.

The resulting adjusted and merged components are then subsequently used as feature vector inputs into a bagging ensemble of 48 Nu-Support Vector Regressor (nu-SVR) models using a linear kernel, implemented using the sklearn python library. The user in turn receives cell type deconvolution results specific to the cell type signatures used as a contextualized reference.

## Synthetic mixture generation and spatial downsampling

To assess UCDBase & UCDSelect performance, we generated pseudobulk mixtures using well-characterized baseline scRNA-Seq data collected from multiple tissue types profiled with scRNA-Seq including *human* PBMCs, lung, and retina (see Supplementary Table 1). Additionally, using an approach similar to Li. et al. 2022 in their Spatial Benchmarking paper[27], we computationally downsampled a high-resolution (10 um) mouse hippocampal spatial dataset profiled using Slide-SeqV2[78] and made available in the squidpy python package via the function sq.datasets.slideseqv2 to create synthetic spatial mixture profiles of lower 100 um resolution with known cell type fractions.

For the PBMC dataset, we performed standard scRNA-Seq preprocessing, dimensionality reduction, and clustering, followed by manual cell type annotation using canonical markers, identifying 8 unique cell types (see Supplementary Fig. 7). Preprocessed and annotated lung and retinal tissue datasets were downloaded from the cellxgene database[79]. For each tissue type we selected one of the two paired datasets and generated 500 pseudobulk mixtures of 100 total cells, representing two to ten randomly selected cell types. We note that datasets utilized for mixture generation were not used in the training of UCDBase. The corresponding paired dataset for each tissue was then utilized as a reference profile. Cell types were matched between both reference and mixture datasets such that references and mixtures both contained the same possible cell types, with no outliers.

To further assess UCDBase & UCDSelect performance on bulk-RNA Seq data, we obtained reference and mixture datasets curated by the recent community DREAM challenge focused on bulk-RNA deconvolution[26], and compared deconvolution accuracy to results available from challenge participants.

## Performance evaluation

We deconvolved mixtures using nine competing approaches developed for different deconvolution applications. Comparators primarily developed for spatial deconvolution include Cell2Location, Stereoscope, Tangram, destVI, SPOTlight, and RCTD. Bulk deconvolution approaches utilizing scRNA-Seq references include SCDC, MuSiC, and Scaden.

Each comparator method was run using published default parameters as recommended by vignettes available at the time of writing, unless otherwise stated. Tangram was run in "cluster" mode for improved performance, given we were only evaluating deconvolution. Reference datasets were randomly downsampled to retain 5,000 cells. Because existing deconvolution methods are sensitive to collinearity and most recommend a degree of input gene filtering, input dimensionality for comparators was limited to the top 7,000 most highly variable genes in the source dataset, as determined by the scanpy function sc.tl.highly_variable_genes using the seurat_v3 method.

We measured performance on the basis of how well UCDBase and UCDSelect were able to predict cell type fractions relative to ground truth. We reported results using Lin's concordance correlation coefficient (CCC), a measure similar to pearson's R, but one that is sensitive to both slope and intercept in addition to variance, making it a suitable metric for comparing deconvolution performance.

## Sensitivity analysis

We reasoned that deconvolution performance of UCDBase and UCDSelect would be sensitive to several hyperparameters pertaining to model complexity, notwithstanding the total cells in a bulk sample, the number of unique cell types present, and fraction of gene dropout.

Using our three scRNA-Seq mixture datasets, we established baseline mixing hyperparameters consisting of 100 cells, 5 unique cell types per mixture, and 0% gene dropout. We then systematically perturbed each variable and generated 500 distinct mixtures, followed by deconvolution and performance evaluation. Total cells in a sample varied from 1 to 1000. The number of unique cell types in a sample varied from 1 to 8. Then, we tested the effect of gene dropout by randomly removing between 0 and 100% of all expressed genes in each mixture at the input stage.

## Integrated gradients analysis

Deep neural network (DNN) models are often described as being "black-box" in nature, whereby the underlying mechanisms correlating inputs to outputs are largely unknown. The ability to interpret DNN models is highly desirable in biomedical science, as it enables researchers to verify a model is learning to generate predictions using plausible mechanistic correlations. Furthermore, interpretability can potentially deliver unique insights into biological processes as they pertain to input genes correlating with model outputs such as cell types. Several approaches for DNN interpretability have been proposed, including model agnostic approaches such as Shapley (SHAP) values[80], Local Interpretable Model-agnostic Explanations (LIME)[81], and DNN-specific methods such as Integrated Gradients (IG)[82]. IG differentiates itself from competing approaches with respect to its scalability to large input dimensions, making it particularly appropriate for interpreting UCD predictions with a 28,867 gene input space. While IG is only applicable to fully differentiable models, making it unsuitable for interpretation of ML methods such as gradient boosted trees or random forest, UCD's implementation as a pure DNN makes it fully compatible with integrated gradients. The goal behind IG is calculation

of the effect a change in a particular input $i$ has on a given output class probability $j$, expressed as the gradient (i.e. partial derivative of $j$ with respect to $i$). The integrated component refers to the accumulation (i.e. mathematical integration) of local gradients for input $i$ across an interpolated range of values starting from a zero-baseline to its true value within a particular sample. Integrated gradients for each input gene are then multiplied by a scaling factor representing the absolute difference between the baseline case and normalized sample expression level, such that only genes actually expressed in the sample being analyzed will yield non-zero input attributions. Intuitively, this enables one to attribute the importance of input (gene) $i$ with respect to how much it is adding to (positive attribution) or subtracting from (negative attribution) the models overall output probability for a given class (celltype) $j$. The intuition behind this approach is visualized in Supplementary Fig. 2.

For IG Analysis (IGA) in UCD, our baseline interpolation function consists of a 50-step linear interpolation of gene expression between zero and true sample values, multiplied by randomized gene dropouts (with a 50-step descending probability of 100% to 0% dropout, as a means of roughly simulating the effect of lower-read depth on absolute gene transcript detection). We approximate the integral of interpolated local gradients using a trapezoidal Riemann summation.

### Secondary spatial and Bulk-RNA-Seq data acquisition and preprocessing

We collected five publicly available, temporal spatial transcriptomics datasets from a *mouse* bilateral renal IRI model developed by Dixon et al. 2022[31]. Breast Invasive Adenocarcinoma and Prostate Adenocarcinoma Spatial FFPE samples were downloaded from the 10X Genomics Datasets repository (see Supplementary Table 1). Colorectal ST data was downloaded from the 10X Genomics Datasets repository by means of the scanpy function sc.datasets.visium_sge.

Bulk-RNA Seq lung data originating from 5 mg tissue samples of patients with ALI, IPF, and healthy lungs collected by Sivakumar et al. 2019[83] was downloaded from the Gene Expression Omnibus (GEO) using accession GSE134692. Bulk-RNA Seq data of white matter lesions sampled from patients with multiple sclerosis or healthy controls by Elkjaer et al. 2019[84] was downloaded from GEO using accession GSE138614. Bulk-RNA Seq data from Fadista et al. 2014[85] comprising pancreatic islet samples from individuals with varying states of T2DM was downloaded from GEO using accession GSE50244. Severity of T2D is monitored long-term by the measure of Hemoglobin % A1c (HgA1c). Values <5.7% are considered "Normal", values between 5.7 and 6.4 are considered "Prediabetes" while values >6.4% indicate a patient has T2DM[86]. Samples were stratified by patient HgA1c clinical thresholds into three groups: normal, prediabetes, and diabetes.

For each bulk-RNA-Seq dataset, TMM-normalized (Lung & Pancreas) or raw (MS) count data, gene annotations, and clinical metadata were integrated into a single annotated dataset object. No filtering was performed on genes or read counts, however read depths for raw counts were normalized to 10,000 per sample. Depth-normalized count data was then passed to UCD for deconvolution. Wilcoxon rank-sums test was used to determine differences in deconvolve cell type fractions between groups, with bonferroni correction for multiple testing.

### Primary non-small cell lung cancer data acquisition and preprocessing

Paired biopsies reflecting tumor and matched adjacent normal tissue were obtained from a patient with non-small cell lung cancer (NSCLC) undergoing surgical resection at the Mount Sinai Hospital (MSH) via the Mount Sinai Pathology Core. Samples were dissociated into single-cell suspensions using the Miltenyi Tumor Dissociation Kit

(130-095-929) and the Miltenyi gentleMACS Dissociator (130-093-235). Single cell suspensions were processed with the 10X Genomics Chromium Next GEM Single Cell 3' v3.1 kit (PN-1000121), targeting 10,000 loaded cells per sample. Whole-transcriptome sample libraries were sequenced on a NovaSeq 6000, targeting 50,000 reads per cell. Sequenced data was processed through CellRanger, yielding filtered count matrices for use as input into downstream single-cell data analysis using the python scanpy package. Both count matrices were concatenated into a single merged dataset. Briefly, cells with <2000 or >100,000 reads were filtered out, as well as cells that contained <200 or >30,000 unique genes. Cells with >10% mitochondrial gene fractions were assumed to be dead or damaged, and excluded from further analysis. Cell counts were normalized to 10,000 counts per cell, and subsequently, the effects of total counts, percent mitochondrial counts, and cell cycle score were regressed out. Regressed, normalized counts were then log-scaled and z-scored with a min-max of +/−10. Highly variable genes were identified on the basis of a dispersion score of 0.1 or greater for genes with log-normalized expression values between 0.1 and 20. HVGs were used to generate 75 principal components. At this stage, we performed batch correction using harmony, which outputs a corrected principal components array for use in all subsequent analysis steps. Calculation of nearest neighbors using our adjusted PCA vectors was done with n_neighbors set to 30. UMAP was used for final dimensionality reduction with minimum_distance set to 0.3. Leiden clustering was then performed to identify transcriptionally-related clusters, with resolution set to 1. Log-normalized counts were used as input into UCD to generate cell type prediction scores.

### Statistics and reproducibility

Unless otherwise noted, for all box plots depicted in this study the center line, box limits and box whiskers correspond to the median, first and third quartiles, and the 1.5x interquartile range, respectively. For bar plots, bar heights correspond to the mean value of the population being visualized. Bar heights denote 95% confidence interval (CI). Unless otherwise noted, $p$-values comparing distributions between groups across box or bar plots were calculated using unpaired two-sided Wilcoxon rank sum test, with Benjamini-Hochberg correction for multiple comparisons where appropriate.

For experimental single cell profiling of lung cancer tissue, the two samples were selected for processing and sequencing on the basis of cellular viability and minimal debris post-dissociation. No statistical methods were used to predetermine sample size for benchmarking, and all available samples were used as described and provided in the literature for each study. Reproducibility of the computational analysis presented in this manuscript is achieved through robust benchmarking, and public availability of both datasets and analysis code included in the supplementary software file. All attempts at replication and validation of the results presented were successful.

### Reporting summary

Further information on research design is available in the Nature Portfolio Reporting Summary linked to this article.

## Data availability

Data generated for this manuscript (scRNA-Seq count matrices, used in Fig. 5) has been deposited and is publically available at https://github.com/dchary/ucdeconvolve_paper. All accession codes for publicly available studies used as training data for the model presented in this study are listed in Supplementary Table 5. All previously published datasets used for benchmarking in this study are listed along with web links to source data in Supplementary Table 1, and are additionally made available at https://github.com/dchary/ucdeconvolve_paper. Source data for bar and box plots in this paper are provided with this paper. Source data are provided with this paper.

## Code availability

The UniCell Deconvolve user API is publically available as a Python package and can be found at https://github.com/dchary/ucdeconvolve/. Documentation and tutorials are available at https://ucdeconvolve.readthedocs.io/en/latest/. All jupyter notebooks used to collect, analyze, and visualize results presented in this manuscript, with an associated list of all external software libraries and corresponding versions, are publicly available at https://github.com/dchary/ucdeconvolve_paper, as a supplementary software file, and upon request.

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

## Acknowledgements

Diagrams and illustrations presented in this manuscript were created using BioRender.com.

## Author contributions

D.C. and R.S. conceived the study. D.C. developed and implemented the model with feedback from R.S., and worked on validation studies and analyses for the manuscript. R.B. provided lung tissue samples used for

prospective validation. D.C. prepared single cell libraries with samples provided by R.B. D.C. and R.S. wrote the manuscript, with feedback from R.B.

## Competing interests

D.C., R.B., and R.S. declare no competing interests. RS is a consultant/advisor as of July 2022 at GeneDx, a company with no direct relationships to the present work.
