## [Peer Review File · Nature Communications]

Reviewers' Comments:

Reviewer #1:

Remarks to the Author:

Charytonowicz et al. collected a large scRNA-seq data and developed an interpretable deep learning model, UniCell: Deconvolve (UCD), for deconvolving cell type composition in spatially resolved transcriptomic, or bulk RNA-seq data. By comparing with the existed methods, UCD shows a comparable performance. Then, the authors applied UCD in different tissues, and demonstrated that UCD could capture some expected/known biology.

Because of the low resolution of most sequencing-based ST, deconvolution is a hot topic in this area. Both reference-based (Cell2Location, CARD) and reference-free (STdeconvolve) approaches are developed. This paper is not based on a single scRNA-seq data but on a large scRNA-seq dataset, which may have potential advantages. However, the authors need more analysis to prove the superiorities.

Major comments:

The authors collected 28 million cells covering 840 cell types. This massive amount of data can bring enormous value to the community. However, the data quality should be clearly described in the manuscript.

a) How strong is the batch effect? Harmony was used for batch correcting. The performance should be described in the manuscript. At least, UMAPs before and after Harmony should be provided. Later, the authors used a spherical variational autoencoder (sVAE) for dimensionality reduction and label transfer. The Sup Fig 1e shows sVAE can somehow remove the batch effect. Which is better when compared to Harmony? If the authors believe sVAE is better, the evidence should be provided.

b) How accurate is the label transfer method? How label transfer accuracy affects the UCD deconvolution. If the authors only use the data with cell type annotations available to train the DNN, the result will be better or not?

c) How many cells are from mouse/human?

One weakness of this manuscript is the benchmark, which is only performed on one dataset (Fig 2). More comprehensive benchmarking needs to be done to demonstrate the improvement over other methods.

a) More datasets, both Imaging-based (eg: MERFISH, seqFISH) and sequencing-based (eg: Slide-seq, sterero-seq), should be included for benchmarking. UCD is trained on the 28 million cells, so it is best to benchmark on other datasets than the training data.

b) The author claimed UCD could also work with bulk RNA-seq (Fig 4), which also needs benchmark.

There are both mouse and human in the training data. How to deal with the cross-species analysis?

I suggest performing a Gene Ontology enrichment analysis of the feature attributes found in different cell types, which could further provide biological insights.

UCD also offer a function to assist cell type annotation (Fig 6). It's better to provide a heatmap/dot-plot for gene expression of the markers based on UCD cell annotation.

Data and code should be made available to the public.

Minor comments:

All the data used in this study should be listed in a table.

Gene symbols need italicized. Mouse genes should have first letter capitalized followed by lowercase. Human genes should be all in uppercase.

In some figures, the font size is too small to clearly see. Eg: Fig 2c, 2d, 3h ...

Enlarge the panel size of Fig 3 g, h.

Use uniform case of letters in the manuscript and figures. "t cell" or "T cell", "cd4" or "CD4", GTEX or "GTEX", "1E-5" or "1e-5", "Type ii" or "Type II" et al.

Lowercase letters for labelling panels in figures, but uppercase in manuscript.

Fig. 2c: Method names should be case sensitive. Such as: "music" should be "MuSiC", "csx" should be "CIBERSORTx", et al.

Reviewer #2:

Remarks to the Author:

Charytonowicz et al. present UCD, a model for deconvolving transcriptomics data pretrained on a large collection of scRNA-seq data. Cell type deconvolution analysis is very important for the emerging spatial transcriptomics data that do not have single-cell resolution. Most existing methods relies on paired scRNA-seq data. This work introduces an attractive approach by utilizing the current large collection of scRNA-seq datasets to derive a generally applicable pretrained model without the need of searching for paired scRNA-seq data. The method achieves better performance compared to several existing deconvolution methods in a benchmark data. This tool is timely with a sound concept and has a good potential for general applicability to the fastly growing spatial transcriptomics.

1. The quality and properties of different scRNA-seq technologies such as Smart-seq and Dropseq are different. However, a uniform preprocessing pipeline is used. The validity of this uniform approach should be demonstrated with detailed real examples.
2. sVAE was used to transfer labels to unannotated single-cell data. Since the quality of annotation is crucial to the following deconvolution step, this label transfer step should be validated both in terms of accuracy and robustness. An overall quality statistic for all the labels assigned to unannotated data should be provided.
3. While this approach using pretrained model without any reference data is universal, it might cause problem if some cell types in a new sample (e.g., a spatial transcriptomics data) are poorly represented in the training data. I thus suggest that a confidence score for each deconvolved spot/sample to be provided to users.
4. The model architecture seems standard and similar to existing approaches. Could the authors clarify what treatments are needed to handle such huge training data? This will be helpful for the future large-scale studies.
5. Related to point 3, will it be helpful to provide a utility to further tune the pretrained model given paired scRNA-seq data? Implementing this is perhaps beyond the scope of this paper but I think this point is worth mentioning in the Discussion.
6. UCD is only quantitatively compared to other methods on PBMC data. One of the advantages of UCD is its general applicability. More benchmarks should be added.
7. UCD uses all possible references, it is interesting to see the comparison to some methods at the other extreme, e.g., STdeconvolve which is reference free.
8. The applicability to spatial transcriptomics technology in addition to Visium should be demonstrated, for example, Slide-seq.

Reviewer #3:

Remarks to the Author:

Cobos and colleagues build a fully-connected neural network to deconvolve cell type fractions. The model is trained on a combination of 899 published single-cell datasets. The authors demonstrate its application in cell type devolution tasks for scRNA-seq, spatial transcriptomic and bulk RNA-seq datasets.

It is an interesting idea to pretrain the model on a huge amount of published data, so that the model does not require specific reference expression profiles when used in downstream tasks — a limitation for most deconvolution algorithms on the market. The authors carry out a comprehensive search for available single-cell datasets. The workflow and model architectures are well described. The authors demonstrate the model's performance on a variety of applications. The results are well organized with adequate interpretations.

Overall, the study addresses an important problem in cell type deconvolution and proposes a promising method. There are a few issues that need to be addressed.

1. Benchmark the performance of UCD for spatial transcriptomics.

The concern is that UCD is only trained on scRNA-seq datasets and how well it performs on spatial transcriptomics remains unclear. In the current case study of spatial transcriptomics, no quantitative evaluation metric is provided and there lacks comparison with other deconvolution algorithms developed for spatial transcriptomics. As spatial transcriptomics is one important use case for UCD, It would be helpful to provide some quantitative benchmark.

2. In the benchmark of UCD for scRNA-seq mixtures, there is some concern about the validity of benchmark dataset, choice of comparison methods and data preprocessing.

In the benchmark task for deconvolving cell types in scRNA-seq mixtures, the best-performers are cell2location (method developed for spatial transcriptomics) and csx (method developed for bulk RNA-seq), better than MuSiC (method developed for scRNA-seq reference and reported as one of state-of-the-art methods). Should we interpret this as a general conclusion, or specific to this particular task?

For deconvolution methods that use scRNA-seq data as reference, previous study (Cobos et al. 2020) has shown that DWLC, MuSiC and SCDC are strong candidates. Is there any particular reason why the authors do not use them as comparison (only MuSiC is used), but include methods that were developed for spatial transcriptomics?

The data transformation and preprocessing have a strong impact on the performance of different deconvolution methods (Cobos et al. 2020). In the paper, the related information is "We performed standard scRNA-Seq preprocessing, dimensionality reduction, and clustering". Is it the same as the preprocessing for training data? If so, the normalization per-sample is not the common practice for transcriptomics data analysis, and it would be helpful to discuss how it affects the performance for other methods. If not, it would be helpful to state clearly what are the detailed data transformation and preprocessing methods.

Reference: Avila Cobos, et al. "Benchmarking of cell type deconvolution pipelines for transcriptomics data." Nature communications (2020).

3. Some questions about the model training

- Is there any hyper-parameter search for the neural network?
- Is there any early-stopping mechanism? From Figure 1e, it seems 50 epoches is not the saturation point, what is the consideration behind the choice of 50 epochs?

4. It would be good if the authors could provide a supplementary table containing the list of 899 datasets.

NCOMMS-22-32274

Peer Review Comments & Responses

Introduction

We would like to extend our gratitude to all the Reviewers for their constructive comments and feedback. The questions and recommendations prompted us to carry out additional analysis, validation and development work which we believe has contributed to a substantial improvement in both our method and manuscript. We would like to summarize some of the major changes made to the revised manuscript:

- **The development and addition of a transfer learning mechanism**, *UniCell Deconvolve Select (UCDSelect)* that leverages our context-free model (UCDBase) as a global cell state feature extractor to enable deconvolution using a user-specified cell state signature. We demonstrate that UCDSelect achieves significant performance improvements over existing approaches, driven in large by the robustness of using UCDBase as a feature extraction method.
- Significantly **improved benchmarking** utilizing a wider variety of datasets and technologies, with additional comparator methods.
 - We benchmarked our deconvolution approach on native Spatial Transcriptomics data from Slide-SeqV2 and other *in-situ* methods (STARMap) in addition to 10X Visium. We furthermore added two additional synthetic mixture datasets from human lung and human retina tissues to diversify cell types in our benchmarks.
 - We compare UCDBase and UCDSelect performance against the recent DREAM bulk deconvolution challenge dataset in relation to competing submissions to benchmark bulk-RNA-Seq deconvolution performance.
 - We added additional comparators to our benchmarks, namely *SCDC*, *SPOTlight*, & *RCTD*.
 - Benchmarking against similar data modalities profiled using different technology platforms (i.e. Smart-Seq vs. Seq-Well vs. 10X Chromium) was performed to test the robustness of our model against technology bias
- Comprehensive evaluation of training dataset composition including **technology platform** and **species** origins.

Response Format

Reviewers' comments are addressed point-by-point below.

All **responses to comments are denoted in green text**, with **changes made in the revised manuscript are identified in blue text with corresponding page and line numbers**.

Modified or additional figures generated in response to a Reviewer comment are included in point-by-point responses where feasible.

Excerpts from revised text are shown in purple text.

Text that represents paraphrased information from a related question posed by multiple Reviewers *is noted and written in italic text*.

Reviewer 1

Original Remarks to the Author

Charytonowicz et al. collected a large scRNA-seq data and developed an interpretable deep learning model, UniCell: Deconvolve (UCD), for deconvolving cell type composition in spatially resolved transcriptomic, or bulk RNA-seq data. By comparing with the existing methods, UCD shows a comparable performance. Then, the authors applied UCD in different tissues, and demonstrated that UCD could capture some expected/known biology.

Because of the low resolution of most sequencing-based ST, deconvolution is a hot topic in this area. Both reference-based (Cell2Location, CARD) and reference-free (STdeconvolve) approaches are developed. This paper is not based on a single scRNA-seq data but on a large scRNA-seq dataset, which may have potential advantages. However, the authors need more analysis to prove the superiorities.

Overall Response

We thank Reviewer 1 for their suggestions and agree that both high- and low-resolution data deconvolution is a significant field especially for context-free approaches that versatily leverage deep single cell data to resolve lower resolution or targeted feature-based ST data. We think the following corrections and additions greatly improve these areas of the manuscript and appreciate your consideration.

Point-by-Point Responses

Major Comments

1. How strong is the batch effect? Harmony was used for batch correcting. The performance should be described in the manuscript. At least, UMAPs before and after Harmony should be provided. Later, the authors used a spherical variational autoencoder (sVAE) for dimensionality reduction and label transfer. The Sup Fig 1e shows sVAE can somehow remove the batch effect. Which is better when compared to Harmony? If the authors believe sVAE is better, the evidence should be provided.

We thank the Reviewer for pointing out important questions relating to our use of sVAE, and the relation to Harmony and batch correction as well as label transfer. The question above highlights a lack of clarity in our methods that we would like to expand upon. **No batch correction was performed on individual gene expression profiles for cells used in mixture generation.** The use of harmony, which has been shown by *Korunsky et. al. 2019¹* to effectively account for both biological and experimental batch effects, for the purposes of batch correction, *was done solely in the context of dataset-level visualization to facilitate manual error-checking* and quality control validation of dataset cell type annotations, but was only used in cases where multiple samples were collected for the same dataset (i.e. multiple patients). The batch correction effect shown in the *original Supplementary Figure 1E-b* was a byproduct of training the SVAE model used as the first automated step in coarse label transfer, but we stress that the outputs of the SVAE were not utilized in any sort of batch correction for the purposes of cleaning or modifying the training mixture input data. Given the discussion in the manuscript on batch correction using harmony for the purposes of visualizing datasets during downstream quality-control, we believe that **Supplementary Figure 1E-b** added unwarranted confusion into the methods of the manuscript, and as such we have elected to remove it from the revised manuscript. We have modified the language in the revised **methods** section, **Cell Type Annotation & Label Transfer** (Lines 262 - 298) section to better clarify the role of SVAE and harmony as tools in our data processing strategy.

¹ Korsunsky I, Millard N, Fan J, Slowikowski K, Zhang F, Wei K, et al. Fast, sensitive and accurate integration of single-cell data with Harmony. Nat Methods. 2019;16: 1289–1296.

2. How accurate is the label transfer method?

We thank the Reviewer for this important question, and would like to clarify the strategy for label transfer, as we recognize the original wording in the submitted manuscript may have been unclear about the results of automated label transfer being utilized directly to generate cell type labels for unannotated datasets. The coarse label transfer using SVAE was done in tandem with human-level validation steps. We recognized early on that attempting to perform fully automated label transfer was infeasible, given numerous constraints notwithstanding the fact that many datasets without annotations certainly would contain cell types not found in the annotated subset. We did however, believe that automated transfer of coarse cell type labels (i.e. epithelial, immune, stromal, etc.) would speed the rate at which a human spot-check could be performed on each dataset. In our label transfer process, we want to stress that for each dataset, we manually verified that cell type annotations present in the original publication were represented in our reprocessed data, which in most cases required manual correction of cell type clusters.

To better support the accuracy and validity of our cell type annotations, log-normalized expression profiles of all cells in quality control passing studies were averaged across common cell types between studies, and the top 50 differentially expressed genes for each cell subtype were computed and made available in **Supplementary Table 3**. As a demonstrative example, we highlight a few examples of the top 5 markers for a subset of cell types contained in our database:

type b pancreatic cell	type i pneumocyte	type ii cell of adrenal cortex	type ii pneumocyte
G6PC2	GGTLC1	CYP11B1	SFTPA1
INS	MS4A15	HSD3B1	SFTPA2
UCN3	AGER	APOC4	ROS1
SLC30A8	GPRC5D	CTXN3	SFTPC
KCNK16	GGTLC3	STAR	SLC22A31
...

Visual inspection of top marker genes for a number of well characterized cell types and cross-referencing with known canonical markers provides empirical evidence of accurate cell type label assignments. We have modified the language in the revised **Methods** section, **Cell Type Annotation & Label Transfer** (Lines 262 - 298) to better illustrate this process.

3. How does label transfer accuracy affect the UCD deconvolution. If the authors only use the data with cell type annotations available to train the DNN, will the result be better or not?

We thank the Reviewer for their question, and want to stress that all datasets that required label transfer went through the process as detailed in Response Points 1 & 2, involving human curation and cross-reference validation with the underlying study / publication. We curated annotations within datasets to the level of detail shown in the original corresponding publication, when available. In many cases, the annotations we curated provided of higher quality than those available in the original study (i.e. original population was labeled *lymphocytes* as opposed to *cd4/8+ t cells*). Based on the labeled marker genes shown in Supplementary Table 3 (see excerpt of table in point above), we do not expect the accuracy of any labels within the dataset to be of measurably lower quality than the median user-annotated dataset.

A direct comparison between the performance of UCDBase using only datasets with available annotations and all datasets following curation would be difficult to compare. The search space of the pre-annotated training model would be considerably reduced, as the diversity of cell types coming from pre-annotated datasets is smaller than all the cell types that we were able to identify using manual curation. As one of the major benefits of UCBase is the breadth of representative data, utilizing only pre-annotated data would significantly limit the potential cell types that could be deconvolved.

With that being said, we would expect performance of UCDBase trained using only pre-annotated data to be *marginally reduced* for more common cell types (i.e *b cells*) and *significantly reduced* for rarer cell types, where the benefits of additional data integration to capture diversity of gene expression between cells of a similar type helps to bolster model robustness. It has been suggested by *Cobos et al. 2020* that *the choice of reference profile outweighs the deconvolution technique* in terms of overall deconvolution performance². By maintaining a heterogeneous set of global reference expression profiles from as many cells as possible, this has the effect of improving overall model performance and reliability compared with relying on pre-annotated data alone.

For cases in which the user expects a cell type to be present that is not found in our curated database, we have implemented UCDSlect, a transfer learning algorithm that leverages the weights from pre-trained UCDBase in conjunction with user-defined cell type signatures to provide contextual deconvolution (see response to Reviewer #2 Point #5 for a detailed description of this additional capability)

² Avila Cobos F, Alquicira-Hernandez J, Powell JE, Mestdagh P, De Preter K. Benchmarking of cell type deconvolution pipelines for transcriptomics data. Nat Commun. 2020;11: 5650

4. How many cells are from mouse/human?

The Reviewer highlighted the need to further illustrate the metadata pertaining to our training dataset. We agree that this is extremely valuable to the audience, and we have now included an additional panel in revised **Supplementary Figure 2F** (attached below) whereby we visualize species metadata in relation to the total number of cells represented in the dataset. We stress that this metadata was collected via keyword text-scanning of protocol and summary descriptions available for each accession from public repositories. As such, instances where multiple species keywords were identified are denoted with semi-colons separating each species. In this context, we determined our dataset contains an approximately 2:1 ratio of verified pure human to pure mouse cells based on this analysis.

We perform a more detailed analysis of species representation in the **Methods** section **Training Data Composition & Sensitivity Across Selected Technology Platforms** (Lines 552 - 577). We report the relevant excerpt from the revised text below:

“Looking at species origins, human derived data made up the majority of cells in our database at 43.2%, with 22.1% coming from mice, and 33.1% of cells coming from datasets where both Human and Mouse keywords were found. By and large the vast majority of single cell data in our training dataset (98.4%) is derived from either Human or Mouse sources, which represent the most common species subject to single cell analysis (see **Supplementary Figure 2F-b**).”

Supplementary Figure 2F. Overview of Platform Technology & Species Metadata Among Training Data. **a)** Barplot of platform labels from automated text scanning of project accessions in relation to number of cells represented by log scale. Pie charts (right) shown for linear comparison. **b)** Barplot of species labels from automated text scanning of project accessions in relation to number of cells represented by log scale. Semi-colon indicates multiple species keywords extracted per project. Pie chart (right) shown for linear comparison.

5. There are both mouse and human cells in the training data. How to deal with the cross-species analysis?

We thank the Reviewer for the insightful comment, and recognize the importance of species-specific differences in gene expression patterns among similar cell types. *Baron et. al 2016*³ profiled gene expression differences among pancreatic cells between mice and humans, finding modest positive correlations in expression across species for similar cell types, driven predominantly by canonical marker genes. For example, *Baron et. al demonstrated* that human and mouse beta cells exhibited a positive correlation with an R-squared of 0.42. To determine correlation between mouse and human cells in our training data, we averaged gene expression profiles across studies by cell type, and calculated correlation statistics between gene expression profiles for 235 cell types matched between both species.

Empirically, we saw that for example, human and mouse beta cells in our dataset exhibited a positive correlation with an R-squared of 0.56 (see **plot above**), comparable to the results seen by *Baron et. al*. We saw an overall correlation of R-squared = 0.46 when averaging values across all 235 cell types, and note that stromal and immune cell types appear to exhibit higher degrees of correlation. We have included the full comparative table of correlations between mice and humans in **Supplementary Table 4** (see below for excerpt).

celltype	rsquared	pearsonr	pvalue
mesenchymal cell	0.766547139	0.875526778	0
fibroblast	0.741143568	0.860896956	0
...
muscle cell	0.379038871	0.615661328	0
trophectodermal cell	0.000350932	0.018733177	0.001457763

With respect to how species-specific differences in gene expression contribute to the accuracy of the model, we contend that the relatively balanced representation of at least mouse and human cells injects a degree of species-specific regularization into the training dataset and subsequently the UniCell model to support cross-species analysis. We recognize that performance of the current model may vary across datasets with less representation within the training dataset, and we have revised the manuscript to note these species-specific considerations in the **Methods** section **Training Data Composition & Sensitivity Across Selected Technology Platforms** (Lines 552 - 577). We report the relevant excerpt from the revised text below:

“Looking at species origins, human derived data made up the majority of cells in our database at 43.2%, with 22.1% coming from mice, and 33.1% of cells coming from datasets where both Human and Mouse keywords were found. By and large the vast majority of single cell data in our training dataset (98.4%) is derived from either Human or Mouse sources, which represent the most common species subject to single cell analysis (see **Supplementary Figure 2F-b**). Across all matched cell types, the average correlation between gene expression across mouse and human data was moderately positive (pearsonr $r = 0.46$) (see **Supplementary Table 4**). Given the potential discrepancies in gene expression between species, we therefore suggest that users bear in mind the species of origin when utilizing UCDBase given the species composition underpinning its training dataset.”

³ Baron M, Veres A, Wolock SL, Faust AL, Gaujoux R, Vetere A, et al. A Single-Cell Transcriptomic Map of the Human and Mouse Pancreas Reveals Inter- and Intra-cell Population Structure. *Cell Syst.* 2016;3: 346–360.e4.

6. One weakness of this manuscript is the benchmark, which is only performed on one dataset (Fig 2). More comprehensive benchmarking needs to be done to demonstrate the improvement over other methods.

We thank the Reviewer for their feedback, and agree that additional benchmarking is warranted to more rigorously validate UCD performance. To address this, we added additional methods as comparators (SCDC, SPOTlight, & RCTD) and diversified the tissue of origin variable for synthetic mixture datasets derived from scRNA-Seq data, incorporating two additional datasets comprising human lung and human retina samples obtained from Wang et. al 2020⁴ (lung), Travaglini et. al 2020⁵ (lung), and Cowan et al. 2020⁶ (retina). The results of these benchmarks are reported in revised **Figure 2** (attached below) along with **Supplementary Figure 2B** (attached below). Additionally, we report highly detailed results of these additional mixture benchmarks in the revised **results** section **Single-Cell RNA-Seq Simulated Mixture Data** (Lines 469 - 502) with the relevant excerpt shown below. Note that the analysis described below highlights the complementary nature of our transfer learning utility, UCDSlect, the details of which are described in *Reviewer #2 Response #5*.

“For PBMCs, our pre-trained UCDBase model obtained strong concordance correlation coefficient (CCC) values of 0.816 averaged across the eight cell types identified in our dataset, while UCDSlect achieved CCC of 0.864, 0.921 and 0.92 for deconvolution utilizing gene features only, embeddings only, and both sources, respectively. UCDBase performed comparably with current State of the Art methods such as Cell2Location (C2L) (see **Figure 2b top**), despite the fact that C2L and competing algorithms were trained to exclusively consider the deconvolution of PBMCs. We note that in the PBMC task, the cell type categories used for comparison are distinct and well-defined, indicating that the corresponding cell types found in UCDBase training dataset are likely to be well-aligned with the labels assigned for this task. UCDSlect exhibited superior performance in this benchmarking task compared with all competing methods.

Results seen in Lung and Retina data highlight the importance of accounting for mismatch between UCDBase and target cell type annotations, and the relevance of UCDSlect as a transfer learning extension of UCDBase. We show that preliminary results indicated average concordance (CCC = 0.524 for Retina, CCC = 0.532 for Lung) with high variance when directly comparing annotated cell types from reference data with the corresponding cell types found in UCDBase’s 840 cell type output. We investigated these discrepancies in Supplementary Figure 2B, where we identified cell types with low initial concordance measurements in both Lung and Retina datasets (see **Supplementary Figure 2B-a/c**). We select three low-performing cell types and performed cross-correlation with output vectors of all 840 UCDBase cell types, and plot pearson correlation between the ground-truth labeled cell type and top 16 highest correlated UCDBase outputs (see **Supplementary Figure 2B-b/d**). The results strongly illustrate that UCDBase correctly identifies cellular state identity, albeit the annotation matched within UCDBase does not always perfectly align with those in the target dataset. For example in our lung mixture dataset, “endothelial cells”, which show a direct label matched correlation of effectively zero, are identified by UCDBase as correlating most closely with “lung endothelial cells” (pearson’s R = 0.851). Similar patterns are seen among other examined cell types, supporting the notion that UCDBase is correctly identifying cell types, however label mismatches make it difficult to discern true accuracy when working with benchmarking datasets relying on potentially flawed, user-defined cell types as ground truth labels. It further highlights the importance of detailed interpretation when analyzing the results of a global pre-trained deconvolution model.”

⁴ Wang A, Chiou J, Poirion OB, Buchanan J, Valdez MJ, Verheyden JM, et al. Single-cell multiomic profiling of human lungs reveals cell-type-specific and age-dynamic control of SARS-CoV2 host genes. *Elife*. 2020;9. doi:10.7554/eLife.62522

⁵ Travaglini KJ, Nabhan AN, Penland L, Sinha R, Gillich A, Sit RV, et al. A molecular cell atlas of the human lung from single-cell RNA sequencing. *Nature*. 2020;587: 619–625.

⁶ Cowan CS, Renner M, De Gennaro M, Gross-Scherf B, Goldblum D, Hou Y, et al. Cell Types of the Human Retina and Its Organoids at Single-Cell Resolution. *Cell*. 2020;182: 1623–1640.e34.

Figure 2. Benchmarking UniCell Deconvolution Performance. A) UMAP reduction of peripheral blood mononuclear cells (PBMCs). B) Deconvolution performance across all cell types for 500 pseudo-mixtures of 5 unique cell types per mixture, as measured by concordance correlation coefficient (CCC). C) UMAP reduction of human lung cells. D) Deconvolution performance for 500 pseudo-mixtures of 2-10 lung cell types per mixture, measured with CCC. E) UMAP reduction of human retina periphery cells. F) Deconvolution performance for 500 pseudo-mixtures of 2-10 retinal cell types per mixture, measured with CCC. G) Spatial map of murine hippocampal formation profiled using Slide-SeqV2 colored by cell type. H) Direct comparison of mixture cell type proportions (left) to UCDBase (center) and UCDSlect (right) matched predictions on aggregated low-resolution spatial zones. I) Deconvolution performance of aggregated low-resolution spatial zones across Spatial Transcriptomics methods.

Supplementary Figure 2B. Assessing Label Mismatch Between Target Datasets and UCDBase Annotations a) Raw unpropagated UCDBase prediction results measured by pearsonr and CCC are plotted as barplots for each cell type as provided in the lung reference dataset. b) Top 16 cell type correlations identified between reference dataset target annotation and UCDBase output prediction for lung reference dataset. c) Raw unpropagated UCDBase prediction results measured by pearsonr and CCC are plotted as barplots for each cell type as provided in the retina reference dataset. d) Top 16 cell type correlations identified between reference dataset target annotation and UCDBase output prediction for retina reference dataset.

7. More datasets, both Imaging-based (eg: MERFISH, seqFISH) and sequencing-based (eg: Slide-seq, Stereo-seq), should be included for benchmarking. UCD is trained on the 28 million cells, so it is best to benchmark on other datasets than the training data.

While we originally intentionally focused on ST datasets derived using whole transcriptome sequencing, we agree that the addition of image-based approaches and targeted in-situ approaches empowers the manuscript, and appreciate the suggestion. To address this, we have added both types of Spatial benchmarks, in concert with additional comparator methods to the manuscript. Specifically, we have added a quantitative spatial transcriptomics benchmark, utilizing a Slide-SeqV2 dataset of the murine hippocampal formation from *Stickels et. al 2020*⁷. Using this single-cell resolution (~10 um) dataset with annotated single cell clusters, we downsampled the spatial resolution, integrating gene expression profiles to generate low-resolution spatial mixtures with ground truth cell type proportions, an approach utilized the recent Spatial Transcriptomics benchmarking study “Benchmarking spatial and single-cell transcriptomics integration methods for transcript distribution prediction and cell type deconvolution” published in Nature Methods by *Li et. al. 2022*⁸. The results of these benchmarks are reported in revised **Figure 2** (attached at end of the response to comment #6 above for figure). Additionally, we report results of these benchmarks in the revised **Results** section **Downsampled Spatial Transcriptomic Data** (Lines 513 - 522). An excerpt of these results is shown below:

“We measured the performance of UCDBase and UCDSselect in deconvolution of downsampled mouse hippocampal Slide-SeqV2 spatial transcriptomic data (see **Figure 2g**). We highlight strong visual concordance between three representative ground truth hippocampal cell type annotations and UCDBase / UCDSselect predictions in **Figure 2h**. To quantify performance, we deconvolve downsampled mixtures using several comparator methods developed for spatial transcriptomics, and show that UCDSselect exhibits comparable deconvolution performance relative to state-of-the-art reference-based approaches, with average CCC values of 0.473, 0.494, and 0.519 for features, embeddings, and both sources, respectively (see **Figure 2i**). Tangram and Stereoscope showed the most consistent performance on this dataset, with average CCC of 0.573 and 0.579, respectively.”

We further leveraged the publicly available benchmarking datasets released by Li et. al. to compare UCD performance to existing methods on low-coverage in-situ STARMap Spatial Transcriptomic data. The results of this benchmark are highlighted in **Supplementary Figure 2E**. We report on the results of this important *in-situ* benchmark in the revised **results** section **Hyperparameter Sensitivity Analysis** (Lines 530 - 551). An excerpt of these results and important implications is shown below (underlined elements are shown for emphasis):

“When perturbing gene dropout, we found that significant performance reductions were seen only after >80% of expressed genes in the benchmarking mixture samples were removed as inputs. This robustness to dropout suggests that UCDBase leverages nonlinear combinations of gene sets as the basis of cell type fraction predictions, and is resilient to the noise seen in transcriptomic data, especially at lower read depth. It nevertheless suggested that the current UCDBase architecture may not be appropriately tuned for use with technologies profiling smaller numbers of genes.”

To validate this important distinction, we obtained mixture and reference signatures generated by Li. et. al. 2022 derived from the mouse visual cortex using the in-situ STARmap spatial transcriptomic technology (see Supplementary Figure 2E-a/b). With an input of just 881 genes, we

⁷ Stickels RR, Murray E, Kumar P, Li J, Marshall JL, Di Bella DJ, et al. Highly sensitive spatial transcriptomics at near-cellular resolution with Slide-seqV2. Nat Biotechnol. 2021;39: 313–319.

⁸ Li B, Zhang W, Guo C, Xu H, Li L, Fang M, et al. Benchmarking spatial and single-cell transcriptomics integration methods for transcript distribution prediction and cell type deconvolution. Nat Methods. 2022;19: 662–670.

reasoned that UCDBase performance would be limited by such a degree of sparsity (~97%) relative to the whole transcriptome input space it was trained on.

Unsurprisingly, we see that UCDSelect achieves only modest deconvolution performance (CCC = 0.658, 0.567, and 0.64 for features, embeddings and both sources, respectively). Notably, results indicate that in this scenario, gene expression features, as opposed to UCDBase extracted embeddings, provide superior deconvolution performance (see **Supplementary Figure 2E-c**). We therefore suggest to users that UCD be utilized primarily in cases where whole transcriptome data is available so as to maximize accuracy and performance. “

Supplementary Figure 2E. Unicell deconvolution performance on in-situ STARMap data. A) Spatial scatterplot of annotated cell types of mouse visual cortex. **B)** Representative distributions of cell type frequencies of spatially downsampled STARMap data. **C)** Deconvolution performance of spatial transcriptomics methods against downsampled STARMap data.

8. The author claimed UCD could also work with bulk RNA-seq (Fig 4), which also needs a benchmark.

To address the UCD bulk deconvolution capability, we have included an additional benchmark in which we leveraged the recent DREAM challenge study from *White et al. 2022*, a **Community assessment of methods to deconvolve cellular composition from bulk gene expression**⁹. This study generated a comprehensive benchmarking dataset consisting of mixtures of FACS-sorted immune cell populations profiled using bulk RNA-Seq, with corresponding standardized reference profiles. We profiled these datasets using both UCDBase and UCDSselect, and highlighted performance of our approach in the revised **Supplementary Figure 2C**. We demonstrate that both UCD models perform favorably using whole-transcriptome bulk-RNA-Seq reference and mixture data, placing UCD among top performing bulk deconvolution methods within the context of this benchmarking study, when compared against the results presented from each comparator method in the *White et al.* publication. The results of this benchmark are highlighted in **Supplementary Figure 2C** (shown below). We report on the results of this important *in-situ* benchmark in the revised **Results** section **Bulk RNA-Seq Benchmarking** (Lines 523 - 529). An excerpt of these results is shown below:

“We compared the performance of UCDBase and UCDSselect in deconvolution of gold-standard bulk RNA-Seq mixture and reference profiles developed for the community DREAM bulk RNA-Seq deconvolution challenge with respect to the results obtained by submitted competitor methods (see **Supplementary Figure 2C**). UCDBase achieved a mean score (measured by pearson’s R) of 0.68, placing it in the top half of submitted solutions. In contrast, UCDSselect achieved mean Pearson’s R scores across all 12 tested cell subtypes of 0.793, 0.892, and 0.903 respectively, scoring considerably higher than competing approaches.”

Supplementary Figure 2C. Bulk RNA Deconvolution DREAM Challenge Benchmark a) Heatmap comparing Pearson correlations of individual immune and stromal cell types across DREAM challenge participants in comparison with UDBase & UCDSselect. b) Boxplots of Pearson correlations for DREAM challenge participants and UCD where each point represents a cell type.

⁹ White BS, de Reyniès A, Newman AM, Waterfall JJ, Lamb A, Petitprez F, et al. Community assessment of methods to deconvolve cellular composition from bulk gene expression. bioRxiv. 2022. p. 2022.06.03.494221. doi:10.1101/2022.06.03.494221

9. I suggest performing a Gene Ontology enrichment analysis of the feature attributes found in different cell types, which could further provide biological insights.

We thank the Reviewer for their insightful suggestion, and have performed a demonstrative gene ontology enrichment analysis, which has been incorporated into the manuscript as **Supplementary Figure 3C** (attached below) and has been summarized in the revised **Results** section **Robust Malignant Subtype Identification & Cancer Feature Attribute Analysis** (Lines 644 - 659). An excerpt of the added text is shown below along with the figure:

“We next asked how exactly the feature weights learned by UCDBase distinguish, at a pan-cancer level, malignant vs. non-malignant epithelial cells. We performed differential “relevance” analysis to identify top gene feature weights that tended to be overrepresented and/or underrepresented as predictors among malignant vs. non-malignant epithelial cells across all cancer subtypes. In total, 1,365 genes were identified to be typically positively correlated with malignant cells, while 821 genes were identified to be positively correlated with normal epithelial cells. Each gene set was then subject to GO_BIOLOGICAL_PROCESS_2021 gene set enrichment analysis using Enrichr (see **Supplementary Figure 3C**).

Of significance, we found inflammatory responses to be among the highest upregulated geneset (adj. $p = 5.4E-5$). Numerous genesets pertaining to signaling pathways including PI3K (adj. $p = 0.022$), ERK1/2 cascade (adj. $p = 0.024$), and the MAPK (adj. $p = 0.016$) cascades were also identified to be significantly upregulated, in addition to angiogenesis (adj. $p = 0.008$). In contrast, normal epithelial cell gene features appear to overwhelmingly favor cell cycle and regulatory machinery, such as regulation of G2/M transition of mitotic cell cycle ($p = 2.57e-12$).

Overall, these results appear to suggest that UCDBase may interpret an epithelial cell as cancerous if it exhibits the simultaneous expression of inflammatory and pro-proliferative signaling pathways.

Further gene set analysis of UCDBase learned representations may yield additional insights into the fundamental biology of cancer and other disease processes.”

Supplementary Figure 3C. Gene Set Enrichment Analysis (GSEA) of Grouped Feature Weights Distinguishes Learned Attributes of Epithelial Malignancy. a) Barplot of GSEA for top learned feature weights across pooled epithelial malignancies with associated network plot (b). c) Barplot of GSEA for top learned feature weights across pooled normal epithelial cell types with associated network plot (d).

10. UCD also offers a function to assist cell type annotation (Fig 6). It's better to provide a heatmap/dot-plot for gene expression of the markers based on UCD cell annotation.

We agree with the Reviewer's suggestion, and have added a dot-plot of gene expression of the markers utilized by UCD to annotate the cell clusters in **Figure 6**. We have included this new figure in the revised manuscript under **Supplementary Figure 5D** (see attached below):

Supplementary Figure 5D. Expression of UCD Marker Genes By Annotated Cell Cluster a) Dotplot grouped by top 5 marker genes identified by UCDBase by cell type cluster.

11. Data and code should be made available to the public.

The integrated UniCell Deconvolve API is now made publicly available at:
github.com/dchary/ucdeconvolve

All scripts utilized for this analysis have been submitted as part of this publication review and have also been deposited on GitHub at the following repository, and will be publically available upon publication:
github.com/dchary/ucdeconvolve_paper

Minor Comments

1. All the data used in this study should be listed in a table.

The extensive list of datasets and sources used in this study to generate the training data are listed in **Supplementary Table 5**.

2. Gene symbols need italicized. Mouse genes should have first letter capitalized followed by lowercase. Human genes should be all in uppercase.

We have revised **Figure 3** to reflect proper gene symbol formatting for mouse genes.

3. In some figures, the font size is too small to clearly see. Eg: Fig 2c, 2d, 3h ...

We have re-formatted figures to increase font sizes, including revised **Figure 2** and **Figure 3** and have also included the high resolution files in the revised submission.

4. Enlarge the panel size of Fig 3 g, h.

The panel sizes of **Figure 3g** and **Figure 3h** have been enlarged.

5. Use uniform case of letters in the manuscript and figures. "t cell" or "T cell", "cd4" or "CD4", GTEX or "GTE_x", "1E-5" or "1e-5", "Type ii" or "Type II" et al.

The case of letters in the manuscript corresponding to cell types and labels has been standardized throughout both the text and figures.

6. Lowercase letters for labeling panels in figures, but uppercase in manuscript.

The case of letters used to label figure panels has been standardized throughout the text and figures.

7. Fig. 2c: Method names should be case sensitive. Such as: "music" should be "MuSiC", "csx" should be "CIBERSORT_x", et al.

Method names have been formatted to be case sensitive.

Reviewer 2

Original Remarks to the Author

Charytonowicz et al. present UCD, a model for deconvolving transcriptomics data pretrained on a large collection of scRNA-seq data. Cell type deconvolution analysis is very important for the emerging spatial transcriptomics data that do not have single-cell resolution. Most existing methods relies on paired scRNA-seq data. This work introduces an attractive approach by utilizing the current large collection of scRNA-seq datasets to derive a generally applicable pretrained model without the need of searching for paired scRNA-seq data. The method achieves better performance compared to several existing deconvolution methods in a benchmark data. This tool is timely with a sound concept and has a good potential for general applicability to the fastly growing spatial transcriptomics.

Overall Response

We thank Reviewer 2 for their supportive comments and really agree that UCD will enable the community to more expeditiously deconvolve cell types and associated transcriptional features with historic, current and future ST and single cell datasets. We think the following clarifications and additions significantly improve the manuscript for your consideration.

Point-by-Point Responses

1. The quality and properties of different scRNA-seq technologies such as Smart-seq and Dropseq are different. However, a uniform preprocessing pipeline is used. The validity of this uniform approach should be demonstrated with detailed real examples.

We acknowledge the Reviewers' concern surrounding differences in quality / properties of different scRNA-Seq technologies. To demonstrate the validity of this approach, we first profiled the overall distribution of technology platforms as a function of the total cells represented in our training dataset, shown in revised **Supplementary Figure 2E** (see attached):

Supplementary Figure 2F. Overview of Platform Technology & Species Metadata Among Training Data. a) Barplot of platform labels from automated text scanning of project accessions in relation to number of cells represented by log scale. Pie charts (right) shown for linear comparison. b) Barplot of species labels from automated text scanning of project accessions in relation to number of cells represented by log scale. Semi-colon indicates multiple species keywords extracted per project. Pie chart (right) shown for linear comparison.

This profile was generated using keyword matching of protocol and methods summaries provided in accession metadata for each dataset used in our training set. We determine that approximately 66.4% of cells in our training dataset were derived using the 10X Genomics Chromium pipeline. Drop-Seq, for example, represented 3.7% of profiled cells. A crucial differentiator in terms of platform technology is the use of long-read (i.e. Smart-Seq) vs. short read, UMI-based approaches, as these techniques generate data requiring unique preprocessing to allow for adequate comparisons between samples. We determined that 4.9% of the cells in our dataset were derived from long-read sequencing technologies. With that said, one of the major corrections required for this data type aside from depth normalization is transcript-length normalization. We contend that as we collected preprocessed count matrices for each study, we found the majority of count files derived from longread data were already transcript length normalized (i.e. TPM). Given the automated nature of our data scraping pipeline, we recognize that not all datasets may have been pre-normalized before collection. We believe that given the small percentage of cells derived from this data modality, coupled with the fact that cell mixtures are generated in log-scale space, minimize the impact that technology biases may have on downstream performance of the deconvolution tool.

It is however, important to validate this assumption, and as such we benchmarked the performance of UCD deconvolution on mixtures generated from PBMCs derived using a number of single-cell sequencing technologies from a study conducted by *Ding et. al. 2019*¹⁰.

We highlight the performance of this comparison in revised **Supplementary Figure 2G** (see below), where we demonstrate that UCD shows comparable performance in deconvolution accuracy across platform technologies, including Smart-Seq long-read data.

Supplementary Figure 2G. Performance Difference by Technical Platform a) Deconvolution accuracy of base and select models on PBMC subsets by celltype prepared using different single cell platform technologies and sequencing modalities. b) Aggregate performance of PBMC deconvolution by technology platform.

¹⁰ Ding J, Adiconis X, Simmons SK, Kowalczyk MS, Hession CC, Marjanovic ND, et al. Systematic comparative analysis of single cell RNA-sequencing methods. bioRxiv. 2019. p. 632216. doi:10.1101/632216

2. sVAE was used to transfer labels to unannotated single-cell data. Since the quality of annotation is crucial to the following deconvolution step, this label transfer step should be validated both in terms of accuracy and robustness. An overall quality statistic for all the labels assigned to unannotated data should be provided.

We thank the Reviewer for raising this critical question, and we agree that the quality of annotation is crucial to deconvolution, as such accurate annotation of all training data was an essential consideration in the design of UCD. As we have received a nearly identical question from another Reviewer, we have included an excerpt of our response which we believe addresses this concern.

Paraphrased From Response to Reviewer #1 Question 2

We would like to clarify the strategy for label transfer, as we recognize the original wording in the submitted manuscript may have incorrectly implied that the results of automated label transfer were utilized directly to generate cell type labels for unannotated datasets. The coarse label transfer using SVAE was done in tandem with human-level validation steps.

We recognized early on that attempting to perform fully automated label transfer was infeasible, given numerous constraints notwithstanding the fact that many datasets without annotations certainly would contain cell types not found in the annotated subset.

We did however, believe that automated transfer of coarse cell type labels (i.e. epithelial, immune, stromal, etc...) would speed the rate at which a human spot-check could be performed on each dataset. In our label transfer process, we want to stress that for each dataset, we manually verified that cell type annotations present in the original publication were represented in our reprocessed data, which in most cases required manual correction of cell type clusters.

*With that said, to support the accuracy and validity of our cell type annotations, log-normalized expression profiles of all cells in QC passing studies were averaged across common cell types between studies, and the top 50 differentially expressed genes for each cell subtype were computed and made available in **Supplementary Table 3**. We highlight a few examples of the top 5 markers for a subset of cell types contained in our database with an excerpt from the table:*

type b pancreatic cell	type i pneumocyte	type ii cell of adrenal cortex	type ii pneumocyte
G6PC2	GGTLC1	CYP11B1	SFTPA1
INS	MS4A15	HSD3B1	SFTPA2
UCN3	AGER	APOC4	ROS1
SLC30A8	GPRC5D	CTXN3	SFTPC
KCNK16	GGTLC3	STAR	SLC22A31
...

Visual inspection of top marker genes for a number of well characterized cell types and cross-referencing with known canonical markers provides empirical evidence of accurate cell type label

assignments. We have modified the language in the revised **methods** section, **Cell Type Annotation & Label Transfer** (Lines 262 - 298) to better illustrate this process.

- While this approach using a pretrained model without any reference data is universal, it might cause problems if some cell types in a new sample (e.g., a spatial transcriptomics data) are poorly represented in the training data. I thus suggest that a confidence score for each deconvolved spot/sample be provided to users.

We thank the Reviewer for this useful suggestion, and agree that providing a confidence score for each deconvolved sample would be a beneficial addition for users, especially in the case of UCDBase where context-free predictions are generated. To address this suggestion, we propose a technique for measuring confidence intervals that is supported by the existing UCDBase architecture and can be incorporated into our existing software package as an update. As the architecture of UCDBase was trained with Dropout regularization, the presence of dropout layers within the core UCD model opens up the possibility of leveraging *test-time dropout*¹¹ as a technique for generating confidence intervals for deep learning model predictions. Dropout regularization involves zeroing the incoming weights of a random set of neurons within a given network layer, effectively “dropping out” those connections from propagating deeper into the network. During training, this has the effect of regularizing the network and preventing model overfitting, as no one neuron can in effect just “memorize” the data being use during training, forcing the network to learn more robust representations of the underlying data manifold.

ben khalifa, Amine & Frigui, Hichem. (2016). *Multiple Instance Fuzzy Inference Neural Networks*.

During test time, or inference, dropout is usually disabled. However, studies have suggested that by maintaining active dropout during inference, repeated predictions made on the same data inputs can be used to generate a prediction interval that reliably correlates & corresponds to the true variability within that population¹². We highlight below the capability for UCDBase to report 95% confidence intervals (CI) for individual predictions using the dropout method (`n_resamples = 64`) on our PBMC mixture dataset:

As demonstrated, we are confident that we can implement the optional confidence interval feature in a future update to our existing UCDeconvolve public cloud API. This will require the restructuring of our backend prediction service to allow for selective activation / deactivation of dropout during inference time, in addition to numerous performance optimizations to minimize the impact of additional latency resulting from having to repeatedly run inference on a given input expression vector to generate a distribution of predictions profiles sufficient to derive a confidence interval.

¹¹ Cortes-Ciriano I, Bender A. Reliable Prediction Errors for Deep Neural Networks Using Test-Time Dropout. arXiv [cs.LG]. 2019. Available: <http://arxiv.org/abs/1904.06330>

¹² Cortes-Ciriano I, Bender A. Reliable Prediction Errors for Deep Neural Networks Using Test-Time Dropout. arXiv [cs.LG]. 2019. Available: <http://arxiv.org/abs/1904.06330>

4. The model architecture seems standard and similar to existing approaches. Could the authors clarify what treatments are needed to handle such huge training data? This will be helpful for the future large-scale studies.

We thank the Reviewer for their question surrounding the handling of large-scale datasets. When working with such a large dataset, we faced several challenges throughout the development of this project.

One of the first issues encountered had to do with data storage and access, which prompted the development of a cloud-native data storage solution that enabled easy access and querying of collected datasets, as detailed in **Methods** section **Data Storage** (Lines 251 - 261).

Most subsequent issues pertained with memory management, which was especially difficult during the mixture generation stage, as mixtures had to be loaded into local machine memory to facilitate rapid integration. We detail our solution to this process in **Methods** section **Mixture Formation via Rapid Data Integration** (Lines 139 - 155), of which we include a relevant excerpt below:

“The process of pseudobulk sample generation was implemented in python and optimized for high-performance execution using the python numba package [28]. All hyperparameters T, N, and F were precomputed as described above prior to generating mixtures, and cell type array row locations were pre-indexed to avoid repeat searches and improve performance. A total of 10 million pseudobulk mixtures were generated over the course of 18 hours at a rate of 150 mixtures per second using a total corpus of 28 million annotated single cells into a 28,000,000 x 28867 compressed-sparse-row (CSR) matrix, running on a Google Cloud Engine (GCE) n2d-standard-224 virtual machine (VM) instance with 224 vCPU cores and 896GB system RAM.”

We found that our implementation of this large-scale mixture generation as described above scaled well, and will continue to function as the size of our underlying training dataset increases. In order to better illustrate our implementation and support community efforts for large-scale data integration, we have included all relevant code examples for pseudo-bulk mixture generation in the data package to be made available with the manuscript upon publication.

Once mixtures were successfully generated, each was then subsequently serialized into *TfRecord* objects and saved onto GCS cloud storage buckets. TfRecords are binary data storage formats optimized for rapid I/O from cloud-based storage solutions to facilitate efficient batch-based machine learning model training with the *Tensorflow Framework*. Essentially, by maintaining adherence to best practices and native file formats for the *Tensorflow* framework, we were able to leverage the benefits of this package to work with out-of-memory datasets at scale. We have included all relevant code examples for model training in the data package to be made available with the manuscript upon publication.

Overall, the flexibility of utilizing cloud-based computing platforms was instrumental in the rapid iteration and development of this project, and recommend considering these platforms when developing similar projects.

5. Related to point 3, will it be helpful to provide a utility to further tune the pretrained model given paired scRNA-seq data? Implementing this is perhaps beyond the scope of this paper but I think this point is worth mentioning in the Discussion.

We have implemented a feature-complete transfer learning pipeline that allows users to leverage the pre-trained knowledge of UCBase while benefiting from context-specific cell type signatures. We have attached the relevant description of this new feature found in the revised **Methods** section **UCDSelect Model Overview** (Lines 319 - 347), in addition to revised **Figure 1** (see below) which highlights this new architecture:

“UCDBase was designed to support the unbiased deconvolution analyses of novel datasets in instances where a reference signature is unavailable or unclear. However, when available, we recognize that following an initial assessment of the cellular composition within a dataset, it would be beneficial to enable a mechanism for model fine-tuning using a user-specified cell type signature. The contents of the user defined signature could also be determined and supported by the context-free analysis generated by UCDBase. Ideally, this mechanism would work seamlessly with the existing UCBase pipeline, and leverage the pre-trained base model to increase performance with minimal computational overhead. To that end, we propose a transfer learning extension, called UCDSelect (see **Figure 1c**), that enables users to leverage the benefits of a pre-trained foundation model together with the specificity of a user-defined cell type signature.

As input, UCDSelect takes the same expression data representing a cell type mixture as UCDBase, with an added reference expression dataset including corresponding cell type labels. Reference data is then averaged across cell types to generate a mean expression signature for user-specified cell states. The input data from both mixture and reference are then fed into our pre-trained UCDBase model, with outputs consisting of the middle two dense layers of the neural network, of dimensions 4096 and 2048, respectively. The two feature vectors corresponding to the reference signature are then independently subject to dimensionality reduction using non-negative matrix factorization, an approach similar to that employed by the SPOTlight deconvolution algorithm, albeit with the input representing model features rather than raw gene expression values [18]. Each NMF model is fit using the reference, and used in turn to then transform the mixture. We utilize the *Combat* algorithm [43] to perform batch alignment on the resulting NMF components so as to improve distribution concordance between reference and mixture, which has been used successfully in other reference-driven techniques, such as *CIBERSORTx* [12]. We repeat the above decomposition process using feature-selected gene expression values, and generate a final merged set of batch-corrected NMF components. We found that in most cases UCDBase features achieve improved relative deconvolution performance, the integration of both extraction techniques leads to slightly higher overall accuracy with negligible performance degradation.

The resulting adjusted and merged components are then subsequently used as feature vector inputs into a bagging ensemble of 48 Nu-Support Vector Regressor (nu-SVR) models using a linear kernel, implemented using the sklearn python library. The user in turn receives cell type deconvolution results specific to the cell type signatures used as a contextualized reference.”

Figure 1. UniCell Data Collection, Foundation Model Architecture & Transfer Learning Strategy. **a)** (left) Flow chart summarizing data collection strategy. **b)** (left) Training data mixture generation & foundation model architecture. (right) UCD model training performance as measured for different training dataset sizes. **c)** Flow chart summarizing transfer learning model architecture and training strategy.

6. UCD is only quantitatively compared to other methods on PBMC data. One of the advantages of UCD is its general applicability. More benchmarks should be added.

We thank the Reviewer, and concur that additional benchmarking would be beneficial to demonstrate the general applicability of UCD. In order to address this, we have expanded our benchmarking cases, adding two additional synthetic mixture datasets derived from scRNA-Seq, including a pair of human lung datasets obtained from *Wang et al. 2020*¹³ and *Travaglini et al. 2020*¹⁴, as well as a human retina dataset obtained from *Cowan et al. 2020*¹⁵. We highlight the results of these new benchmarks in revised **Figure 2** (see below) along with **Supplementary Figure 2B** (see below). Furthermore, we detail the results of the additional mixture benchmarks in the revised **results** section **Single-Cell RNA-Seq Simulated Mixture Data** (Lines 469 - 512), an excerpt of which is shown below:

“For PBMCs, our pre-trained UCDBase model obtained strong concordance correlation coefficient (CCC) values of 0.816 averaged across the eight cell types identified in our dataset, while UCDSselect achieved CCC of 0.864, 0.921 and 0.92 for deconvolution utilizing gene features only, embeddings only, and both sources, respectively. UCDBase performed comparably with current State of the Art methods such as Cell2Location (C2L) (see **Figure 2b top**), despite the fact that C2L and competing algorithms were trained to exclusively consider the deconvolution of PBMCs. We note that in the PBMC task, the cell type categories used for comparison are distinct and well-defined, indicating that the corresponding cell types found in UCDBase training dataset are likely to be well-aligned with the labels assigned for this task. UCDSselect exhibited superior performance in this benchmarking task compared with all competing methods.

Results seen in Lung and Retina data highlight the importance of accounting for mismatch between UCDBase and target cell type annotations, and the relevance of UCDSselect as a transfer learning extension of UCDBase. We show that preliminary results indicated average concordance (CCC = 0.524 for Retina, CCC = 0.532 for Lung) with high variance when directly comparing annotated cell types from reference data with the corresponding cell types found in UCDBase’s 840 cell type output. We investigated these discrepancies in Supplementary Figure 2B, where we identified cell types with low initial concordance measurements in both Lung and Retina datasets (see **Supplementary Figure 2B-a/c**). We select three low-performing cell types and performed cross-correlation with output vectors of all 840 UCDBase cell types, and plot Pearson correlation between the ground-truth labeled cell type and top 16 highest correlated UCDBase outputs (see **Supplementary Figure 2B-b/d**). The results strongly illustrate that UCDBase correctly identifies cellular state identity, albeit the annotation matched within UCDBase does not always perfectly align with those in the target dataset. For example in our lung mixture dataset, “endothelial cells”, which show a direct label matched correlation of effectively zero, are identified by UCDBase as correlating most closely with “lung endothelial cells” (Pearson’s R = 0.851). Similar patterns are seen among other examined cell types, supporting the notion that UCDBase is correctly identifying cell types, however label mismatches make it difficult to discern true accuracy when working with benchmarking datasets relying on potentially flawed, user-defined cell types as ground truth labels. It further highlights the importance of detailed interpretation when analyzing the results of a global pre-trained deconvolution model.

UCDSselect however, represents a natural extension of UCDBase and a solution to the complexity of label mismatch. By aligning UCDBase’s feature vectors to a user-specified reference signature,

¹³ Wang A, Chiou J, Poirion OB, Buchanan J, Valdez MJ, Verheyden JM, et al. Single-cell multiomic profiling of human lungs reveals cell-type-specific and age-dynamic control of SARS-CoV2 host genes. *Elife*. 2020;9. doi:10.7554/eLife.62522

¹⁴ Travaglini KJ, Nabhan AN, Penland L, Sinha R, Gillich A, Sit RV, et al. A molecular cell atlas of the human lung from single-cell RNA sequencing. *Nature*. 2020;587: 619–625.

¹⁵ Cowan CS, Renner M, De Gennaro M, Gross-Scherf B, Goldblum D, Hou Y, et al. Cell Types of the Human Retina and Its Organoids at Single-Cell Resolution. *Cell*. 2020;182: 1623–1640.e34.

we are effectively able to guide UCD to a solution within the parameter space defined by the user. For the Lung benchmark, UCDSelect achieves average CCC values of 0.832, 0.861, and 0.883 for features, embeddings, and both sources, respectively. The Retina benchmark saw average CCC values for UCDselect of 0.93, 0.97, and 0.972 for features, embeddings, and both sources, respectively. The strong performance on the Retina benchmark is unsurprising, given that unlike the PBMC and Lung datasets which featured mixture and reference data derived from different studies, the paired Retina reference and mixture data sources are both derived from two samples from the same study, which likely minimizes the batch and/or experimental related differences between common cell types in these samples.”

Figure 2. Benchmarking UniCell Deconvolution Performance. A) UMAP reduction of peripheral blood mononuclear cells (PBMCs). B) Deconvolution performance across all cell types for 500 pseudo-mixtures of 5 unique cell types per mixture, as measured by concordance correlation coefficient (CCC). C) UMAP reduction of human lung cells. D) Deconvolution performance for 500 pseudo-mixtures of 2-10 lung cell types per mixture, measured with CCC. E) UMAP reduction of human retina periphery cells. F) Deconvolution performance for 500 pseudo-mixtures of 2-10 retinal cell types per mixture, measured with CCC. G) Spatial map of murine hippocampal formation profiled using Slide-SeqV2 colored by cell type. H) Direct comparison of mixture cell type proportions (left) to UCDBase (center) and UCDSelect (right matched predictions on aggregated low-resolution spatial zones. I) Deconvolution performance of aggregated low-resolution spatial zones across Spatial Transcriptomics methods.

Supplementary Figure 2B. Assessing Label Mismatch Between Target Datasets and UCDBase Annotations a) Raw unpropagated UCDBase prediction results measured by Pearsonr and CCC are plotted as barplots for each cell type as provided in the lung reference dataset. b) Top 16 cell type correlations identified between reference dataset target annotation and UCDBase output prediction for lung reference dataset. c) Raw unpropagated UCDBase prediction results measured by Pearsonr and CCC are plotted as barplots for each cell type as provided in the retina reference dataset. d) Top 16 cell type correlations identified between reference dataset target annotation and UCDBase output prediction for retina reference dataset.

To address the UCD bulk deconvolution capability, we leveraged the DREAM challenge study from *White et al. 2022*, a **Community assessment of methods to deconvolve cellular composition from bulk gene expression**¹⁶ to benchmark UCDBase and UCDBase on bulk-RNA-Seq data. The results of this benchmark are shown in **Supplementary Figure 2C** (see below). An excerpt of the revised **results** section **Bulk RNA-Seq Benchmarking** (Lines 523 - 529) is shown below:

“We compared the performance of UCDBase and UCDBase in deconvolution of gold-standard bulk RNA-Seq mixture and reference profiles developed for the community DREAM bulk RNA-Seq deconvolution challenge with respect to the results obtained by submitted competitor methods (see **Supplementary Figure 2C**). UCDBase achieved a mean score (measured by Pearson’s R) of 0.68,

¹⁶ White BS, de Reyniès A, Newman AM, Waterfall JJ, Lamb A, Petitprez F, et al. Community assessment of methods to deconvolve cellular composition from bulk gene expression. bioRxiv. 2022. p. 2022.06.03.494221. doi:10.1101/2022.06.03.494221

placing it in the top half of submitted solutions. In contrast, UCDSselect achieved mean Pearson's R scores across all 12 tested cell subtypes of 0.793, 0.892, and 0.903 respectively, scoring considerably higher than competing approaches.”

Supplementary Figure 2C. Bulk RNA Deconvolution DREAM Challenge Benchmark a) Heatmap comparing Pearson correlations of individual immune and stromal cell types across DREAM challenge participants in comparison with UDBase & UCDSselect. b) Boxplots of Pearson correlations for DREAM challenge participants and UCD where each point represents a cell type.

7. UCD uses all possible references, it is interesting to see the comparison to some methods at the other extreme, e.g., STdeconvolve which is reference free.

We thank the Reviewer for suggesting STDeconvolve as an additional comparator method. To compare the performance of UCD to STDeconvolve, we first sought to apply STDeconvolve to our lung tissue mixture sample. STDeconvolve leverages Latent Dirichlet Allocation (LDA) as a dimensionality reduction / topic modeling method, which decomposes an input gene signature into a set of K topics, which are in principle meant to correspond to cell types.

One of the challenges of LDA is in the selection of K , which in the case of STDeconvolve being a reference-free method, is unknown at runtime for novel tissues. STDeconvolve leverages a set of heuristics to optimize a selection for K , however in the case of our lung benchmark, we supplied $K = 19$ as that was the ground-truth number of cell types present in our mixture. After running STDeconvolve, we were left with 19 deconvolved topic profiles, in addition to expression signatures for those 19 topics, which were used to attempt to match the deconvolved topics to target cell types. Below is the result of a pearson correlation analysis between gene expression patterns from the resulting 19 topics and 19 cell types from our reference sample:

From these results, it is unclear which topic profile should be used to represent which cell type when interpreting the results of our mixture and reference datasets. Given the uncertainty inherent in selecting the optimal number of cell types, especially outside of the context of a controlled benchmark, and the interpretation of topic profile components that may or may not directly correspond to a single cell type, we found it would be difficult to apply STDeconvolve as a direct benchmark to compare between UCDBase, UCDSlect, and our other comparators.

8. The applicability to spatial transcriptomics technology in addition to Visium should be demonstrated, for example, Slide-seq.

We agree with the Reviewer that additional benchmarks relating to different spatial technologies should be added. As a similar question was posed by Reviewer 1, we have included an excerpt of our response which we believe fully addresses this concern.

Paraphrased From Response to Reviewer #1 Question 7

*We have added a quantitative spatial transcriptomics benchmark, utilizing a Slide-SeqV2 dataset of the murine hippocampal formation from Stickels et. al 2020¹⁷. Using this single-cell resolution (~10 um) dataset with annotated single cell clusters, we downsampled the spatial resolution, integrating gene expression profiles to generate low-resolution spatial mixtures with ground truth cell type proportions, an approach utilized the recent Spatial Transcriptomics benchmarking study “Benchmarking spatial and single-cell transcriptomics integration methods for transcript distribution prediction and cell type deconvolution” published in Nature Methods by Li et. al. 2022¹⁸. The results of these benchmarks are reported in revised **Figure 2** (see response to comment #6 above for figure). Additionally, we report results of these benchmarks in the revised **Results** section **Downsampled Spatial Transcriptomic Data** (Lines 513 - 522). An excerpt of these results is shown below:*

“We measured the performance of UCDBase and UCDSlect in deconvolution of downsampled mouse hippocampal Slide-SeqV2 spatial transcriptomic data (see Figure 2g). We highlight strong visual concordance between three representative ground truth hippocampal cell type annotations and UCDBase / UCDSlect predictions in Figure 2h. To quantify performance, we deconvolve downsampled mixtures using several comparator methods developed for spatial transcriptomics, and show that UCDSlect exhibits comparable deconvolution performance relative to state-of-the-art reference-based approaches, with average CCC values of 0.473, 0.494, and 0.519 for features, embeddings, and both sources, respectively (see Figure 2i). Tangram and Stereoscope showed the most consistent performance on this dataset, with average CCC of 0.573 and 0.579, respectively.”

*We further leveraged the publicly available benchmarking datasets released by Li et. al. to compare UCD performance to existing methods on low-coverage in-situ STARMap Spatial Transcriptomic data. The results of this benchmark are highlighted in **Supplementary Figure 2E**. We report on the results of this important in-situ benchmark in the revised **Results** section **Hyperparameter Sensitivity Analysis** (Lines 537 - 551). An excerpt of these results and important implications is shown below (underlined elements are shown for emphasis):*

“When perturbing gene dropout, we found that significant performance reductions were seen only after >80% of expressed genes in the benchmarking mixture samples were removed as inputs. This robustness to dropout suggests that UCDBase leverages nonlinear combinations of gene sets as the basis of cell type fraction predictions, and is resilient to the noise seen in transcriptomic data, especially at lower read depth. It nevertheless suggested that the current UCDBase architecture may not be appropriately tuned for use with technologies profiling smaller numbers of genes.”

¹⁷ Stickels RR, Murray E, Kumar P, Li J, Marshall JL, Di Bella DJ, et al. Highly sensitive spatial transcriptomics at near-cellular resolution with Slide-seqV2. Nat Biotechnol. 2021;39: 313–319.

¹⁸ Li B, Zhang W, Guo C, Xu H, Li L, Fang M, et al. Benchmarking spatial and single-cell transcriptomics integration methods for transcript distribution prediction and cell type deconvolution. Nat Methods. 2022;19: 662–670.

To validate this important distinction, we obtained mixture and reference signatures generated by Li. et. al. 2022 derived from the mouse visual cortex using the in-situ STARmap spatial transcriptomic technology (see **Supplementary Figure 2E-a/b**). With an input of just 881 genes, we reasoned that UCDBase performance would be limited by such a degree of sparsity (~97%) relative to the whole transcriptome input space it was trained on.

Unsurprisingly, we see that UCDSelect achieves only modest deconvolution performance (CCC = 0.658, 0.567, and 0.64 for features, embeddings and both sources, respectively). Notably, results indicate that in this scenario, gene expression features, as opposed to UCDBase extracted embeddings, provide superior deconvolution performance (see **Supplementary Figure 2E-c**). We therefore suggest to users that UCD be utilized primarily in cases where whole transcriptome data is available so as to maximize accuracy and performance.”

Supplementary Figure 2E. Unicell deconvolution performance on in-situ STARMap data. A) Spatial scatterplot of annotated cell types of mouse visual cortex. **B)** Representative distributions of cell type frequencies of spatially downsampled STARMap data. **C)** Deconvolution performance of spatial transcriptomics methods against downsampled STARMap data.

Reviewer 3

Original Remarks to the Author

Cobos [Charytonowicz] and colleagues build a fully-connected neural network to deconvolve cell type fractions. The model is trained on a combination of 899 published single-cell datasets. The authors demonstrate its application in cell type devolution tasks for scRNA-seq, spatial transcriptomic and bulk RNA-seq datasets.

It is an interesting idea to pretrain the model on a huge amount of published data, so that the model does not require specific reference expression profiles when used in downstream tasks — a limitation for most deconvolution algorithms on the market. The authors carry out a comprehensive search for available single-cell datasets. The workflow and model architectures are well described. The authors demonstrate the model's performance on a variety of applications. The results are well organized with adequate interpretations.

Overall, the study addresses an important problem in cell type deconvolution and proposes a promising method. There are a few issues that need to be addressed.

Overall Response

We thank Reviewer 3 for the encouraging comments and agree that the requirement to provide an initial reference expression profile poses a limitation against most currently published deconvolution algorithms. We believe that the revisions described below your feedback have contributed positively to the manuscript and methodology overall.

Point-by-Point Responses

1. Benchmark the performance of UCD for spatial transcriptomics. The concern is that UCD is only trained on scRNA-seq datasets and how well it performs on spatial transcriptomics remains unclear. In the current case study of spatial transcriptomics, no quantitative evaluation metric is provided and there lacks comparison with other deconvolution algorithms developed for spatial transcriptomics. As spatial transcriptomics is one important use case for UCD, it would be helpful to provide some quantitative benchmark.

We agree with the Reviewer that additional benchmarks relating to different spatial technologies should be added. As a similar question was posed by both Reviewer 1 and Reviewer 2 pertaining to spatial benchmarking, we have included an excerpt of our response which we believe fully addresses this concern.

Paraphrased From Response to Reviewer #1 Question 7

*We have added a quantitative spatial transcriptomics benchmark, utilizing a Slide-SeqV2 dataset of the murine hippocampal formation from Stickels et. al 2020¹⁹. Using this single-cell resolution (~10 um) dataset with annotated single cell clusters, we downsampled the spatial resolution, integrating gene expression profiles to generate low-resolution spatial mixtures with ground truth cell type proportions, an approach utilized the recent Spatial Transcriptomics benchmarking study "Benchmarking spatial and single-cell transcriptomics integration methods for transcript distribution prediction and cell type deconvolution" published in Nature Methods by Li et. al. 2022²⁰. The results of these benchmarks are reported in revised Figure 2 (see below). Additionally, we report results of these benchmarks in the revised Results section **Downsampled Spatial Transcriptomic Data** (Lines 513 - 522). An excerpt of these results is shown below:*

¹⁹ Stickels RR, Murray E, Kumar P, Li J, Marshall JL, Di Bella DJ, et al. Highly sensitive spatial transcriptomics at near-cellular resolution with Slide-seqV2. Nat Biotechnol. 2021;39: 313–319.

²⁰ Li B, Zhang W, Guo C, Xu H, Li L, Fang M, et al. Benchmarking spatial and single-cell transcriptomics integration methods for transcript distribution prediction and cell type deconvolution. Nat Methods. 2022;19: 662–670.

“We measured the performance of UCDBase and UCDSelct in deconvolution of downsampled mouse hippocampal Slide-SeqV2 spatial transcriptomic data (see **Figure 2g**). We highlight strong visual concordance between three representative ground truth hippocampal cell type annotations and UCDBase / UCDSelct predictions in **Figure 2h**. To quantify performance, we deconvolve downsampled mixtures using several comparator methods developed for spatial transcriptomics, and show that UCDSelct exhibits comparable deconvolution performance relative to state-of-the-art reference-based approaches, with average CCC values of 0.473, 0.494, and 0.519 for features, embeddings, and both sources, respectively (see **Figure 2i**). Tangram and Stereoscope showed the most consistent performance on this dataset, with average CCC of 0.573 and 0.579, respectively.”

Figure 2. Benchmarking UniCell Deconvolution Performance. A) UMAP reduction of peripheral blood mononuclear cells (PBMCs). B) Deconvolution performance across all cell types for 500 pseudo-mixtures of 5 unique cell types per mixture, as measured by concordance correlation coefficient (CCC). C) UMAP reduction of human lung cells. D) Deconvolution performance for 500 pseudo-mixtures of 2-10 lung cell types per mixture, measured with CCC. E) UMAP reduction of human retina periphery cells. F) Deconvolution performance for 500 pseudo-mixtures of 2-10 retinal cell types per mixture, measured with CCC. G) Spatial map of murine hippocampal formation profiled using Slide-SeqV2 colored by cell type. H) Direct comparison of mixture cell type proportions (left) to UCDBase (center) and UCDSelct (right) matched predictions on aggregated low-resolution spatial zones. I) Deconvolution performance of aggregated low-resolution spatial zones across Spatial Transcriptomics methods.

We further leveraged the publicly available benchmarking datasets released by Li et. al. to compare UCD performance to existing methods on low-coverage in-situ STARMap Spatial Transcriptomic data. The results of this benchmark are highlighted in **Supplementary Figure 2E**. We report on the results of this important in-situ benchmark in the revised **results** section **Hyperparameter Sensitivity Analysis** (Lines 530 - 551). An excerpt of these results and important implications is shown below (underlined elements are shown for emphasis):

“When perturbing gene dropout, we found that significant performance reductions were seen only after >80% of expressed genes in the benchmarking mixture samples were removed as inputs. This robustness to dropout suggests that UCDBase leverages nonlinear combinations of gene sets as the basis of cell type fraction predictions, and is resilient to the noise seen in transcriptomic data, especially at lower read depth. It nevertheless suggested that the current UCDBase architecture may not be appropriately tuned for use with technologies profiling smaller numbers of genes.

To validate this important distinction, we obtained mixture and reference signatures generated by Li. et. al. 2022 derived from the mouse visual cortex using the in-situ STARmap spatial transcriptomic technology (see **Supplementary Figure 2E-a/b**). With an input of just 881 genes, we reasoned that UCDBase performance would be limited by such a degree of sparsity (~97%) relative to the whole transcriptome input space it was trained on.

Unsurprisingly, we see that UCDSlect achieves only modest deconvolution performance (CCC = 0.658, 0.567, and 0.64 for features, embeddings and both sources, respectively). Notably, results indicate that in this scenario, gene expression features, as opposed to UCDBase extracted embeddings, provide superior deconvolution performance (see **Supplementary Figure 2E-c**). We therefore suggest to users that UCD be utilized primarily in cases where whole transcriptome data is available so as to maximize accuracy and performance.”

Supplementary Figure 2E. Unicell deconvolution performance on in-situ STARMap data. A) Spatial scatterplot of annotated cell types of mouse visual cortex. **B)** Representative distributions of cell type frequencies of spatially downsampled STARMap data. **C)** Deconvolution performance of spatial transcriptomics methods against downsampled STARMap data.

2. In the benchmark task for deconvolving cell types in scRNA-seq mixtures, the best-performers are cell2location (method developed for spatial transcriptomics) and csx (method developed for bulk RNA-seq), better than MuSiC (method developed for scRNA-seq reference and reported as one of state-of-the-art methods). Should we interpret this as a general conclusion, or specific to this particular task?

We would contend that with regards to deconvolution benchmarks, results are highly specific to the dataset in question. As such, we would not attempt to draw general conclusions pertaining to the overall accuracy of a given method with respect to another based on one benchmark. For this reason, we have expanded our benchmarking cases to capture additional heterogeneity in cell types and organ systems, adding two more synthetic mixture datasets from scRNA-Seq, including a pair of human lung datasets obtained from *Wang et. al 2020*²¹ and *Travaglini et. al 2020*²², as well as a human retina dataset obtained from *Cowan et al. 2020*²³. The results of these more comprehensive benchmarks are highlighted in revised **Figure 2** (see above response to point #1). Furthermore, we detail the results of the additional mixture benchmarks in the revised **Results** section **Single-Cell RNA-Seq Simulated Mixture Data** (Lines 469 - 512). We provided a detailed excerpt of the relevant results and interpretation in our response to Reviewer 2 Point #6.

3. For deconvolution methods that use scRNA-seq data as reference, previous study (Cobos et al. 2020) has shown that DWLC, MuSiC and SCDC are strong candidates. Is there any particular reason why the authors do not use them as comparison (only MuSiC is used), but include methods that were developed for spatial transcriptomics?

We thank the Reviewer for their comments and reference to the *Cobos et al.* manuscript. To address these points, we have now included SCDC in addition to MuSiC as per the Reviewer's suggestion as comparator methods. Due to technical constraints, we were unable to successfully evaluate DWLS performance on our benchmarking mixture datasets.

²¹ Wang A, Chiou J, Poirion OB, Buchanan J, Valdez MJ, Verheyden JM, et al. Single-cell multiomic profiling of human lungs reveals cell-type-specific and age-dynamic control of SARS-CoV2 host genes. *Elife*. 2020;9. doi:10.7554/eLife.62522

²² Travaglini KJ, Nabhan AN, Penland L, Sinha R, Gillich A, Sit RV, et al. A molecular cell atlas of the human lung from single-cell RNA sequencing. *Nature*. 2020;587: 619–625.

²³ Cowan CS, Renner M, De Gennaro M, Gross-Scherf B, Goldblum D, Hou Y, et al. Cell Types of the Human Retina and Its Organoids at Single-Cell Resolution. *Cell*. 2020;182: 1623–1640.e34.

4. The data transformation and preprocessing have a strong impact on the performance of different deconvolution methods (Cobos et al. 2020). In the paper, the related information is “We performed standard scRNA-Seq preprocessing, dimensionality reduction, and clustering”. Is it the same as the preprocessing for training data? If so, the normalization per-sample is not the common practice for transcriptomics data analysis, and it would be helpful to discuss how it affects the performance for other methods. If not, it would be helpful to state clearly what are the detailed data transformation and preprocessing methods.

We would like to thank the Reviewer for raising this key point, and agree with the statement that transformation and preprocessing have a strong impact on performance of different deconvolution methods. We first would like to clarify that the statement “*Inputs are normalized on a per-sample basis rather than per-gene, as is standard in transcriptomics data analysis.*” made in **Methods** section “**Primary Data Input & Preprocessing**” (Line 82) referred to “sample” in the context of a 1-dimensional input gene expression, where “sample” would refer to a single cell or mixture of cells. We recognize this wording may incidentally be misconstrued to be referring to the whole dataset (i.e. a scRNA-Seq *sample* of ~2,000 cells) from which that particular cell or mixture came from, in which case we wholly agree that form of normalization would not be appropriate. We have clarified this terminology in the revised **Methods** section “**Primary Data Input & Preprocessing**” (Lines 82 - 89).

Regarding the statement “*We performed standard scRNA-Seq preprocessing, dimensionality reduction, and clustering*” made in the **Methods** section “**Synthetic Mixture Generation**” (Line 356), we would like to clarify what data transformations were performed at distinct stages of the pipeline.

Training Data Collection & Mixture Formation

As part of our data curation efforts, as described in the **Methods** section “**Data Preprocessing**” (Lines 216 - 250) we followed currently accepted best practices²⁴ for scRNA-Seq data preprocessing for each sample (i.e. dataset) we obtained. More specifically, for each cell we ensured that counts were depth (i.e. CP10K) normalized, followed by *log1p* scaling, both transformations which are cell-specific and independent of the effects of other cells in the sample or other samples collected. While *Cobos et. al* reports that, for the purposes of bulk RNA Deconvolution, linear data was found superior to log-transformed, we contend that log transformations in the context of our mixture and deep learning architecture are beneficial. With respect to generation of mixtures derived from numerous datasets originating from a wide range of technology platforms and sequencing parameters, log-transformation stabilizes count variance between cells, thereby reducing the effect of outliers on skewing mean gene expression profiles. On the other hand, deep learning models, and associated weights, are highly sensitive to the dynamic range of input variables, especially for those whose values (i.e gene expression) can span several orders of magnitude. Log scaling genes put all genes in a given cell with the same order of magnitude (1 ~10), supporting more evenly balanced weights across all input features.

Internal Model Data Transformation

All inputs into Unicell Deconvolve internally undergo two additional critical normalization steps. Each input (i.e. a mixture consisting of a vector of gene expression) is z-scored, followed by min-max scaling. Both these transformations further reduce scaling and variance differences between input features, which subsequently promote more balanced model training and weight assignments.

Data Transformation for Benchmarking Comparisons

We recognize that other deconvolution methods, including those compared in the requested study benchmarks, are equally sensitive to the scaling and normalization strategies used on input data. With that said, all synthetic mixtures generated as part of our benchmarking are *only depth normalized*, and are provided as *linear* scale inputs unless otherwise specified by the method’s requirements. We have added an additional subsection into the revised **Methods** section **Performance Evaluation** (Lines 368 - 383) where we specify additional details regarding input parameters and parameter choices for each method.

²⁴ Luecken MD, Theis F.J. Current best practices in single-cell RNA-seq analysis: a tutorial. *Mol Syst Biol.* 2019;15: e8746.

5. Is there any hyper-parameter search for the neural network?

Unicell Deconvolve model architecture was developed after numerous iterations of both manual and semi-automated hyperparameter grid searches in the following steps:

Baseline Architecture Choice

We initially sought to determine the optimal baseline architecture, considering both convolutional (i.e. Renet Inception modules, etc...) and more straightforward fully connected modules, both in isolation and sequentially, in conjunction with varied input formats. Gene expression signatures were initially mapped using protein interaction graphs to a 2-dimensional spatial grid structure, hypothesizing that convolutional module(s) would be able to derive additional information pertaining to cell type identity. We found that complex data inputs and convolutional modules provided significantly lower performance compared with simpler models consisting of sequential dense layers following a 1-dimensional non-sorted expression vector. As such, we opted for a fully dense baseline architecture, as shown in revised Figure 1b (see revised Figure 1 under Reviewer 2 Point #5 response).

Hyperparameter Tuning

After choosing a baseline architecture, we contended with the decision(s) surrounding the number of layers to use, activation function, regularization methods, learning rate, and batch size. Given the amount of training data utilized in the final model, we subset our core training dataset randomly down to 250K - 1M mixtures for the purposes of tuning, and repeatedly trained smaller models out to no more than 10 epochs, looking to see the effect of sequential changes to individual on training performance. As can be seen in revised Figure 1b right (see revised Figure 1 under Reviewer 2 Point #5 response), model performance within the first N epochs is indicative of eventual performance upon convergence regardless of training set size, making this a suitable proxy for subsequent rounds of optimization.

A more detailed description of the neural network architecture and design choices has been incorporated in the revised Methods section “Primary Data Input & Preprocessing” (Lines 73 - 96)

6. Is there any early-stopping mechanism? From Figure 1e, it seems 50 epoches is not the saturation point, what is the consideration behind the choice of 50 epochs?

The Reviewer raises an important question regarding optimization and tuning of the model training process. We leveraged two training callbacks from the *tf.keras.callbacks* module, including *tf.keras.callbacks.EarlyStopping* as well as *tf.keras.callbacks.ReduceLROnPlateau* with the following hyperparameter choices for model training:

Module	Hyperparameter	Value	Interpretation
EarlyStopping	patience	4	If model loss, measured by sparse Mean Squared Error (MSE), does not improve by a delta of 1.25e-5 between any successive four epochs (defined as one full iteration across all batches in training dataset), then model training should terminate.
	monitor	"loss"	
	min_delta	1.25e-5	
ReduceLROnPlateau	patience	5	If model loss does not improve by a delta of 1.0e-4 between any successive five epochs, the the model learning rate should be reduce by a factor of 50% of its current value to improve convergence to a minimum loss over successive epochs.
	monitor	"loss"	
	factor	0.5	
	min_delta	1.0e-4	

We agree with the Reviewer that original Figure 1E (now revised **Figure 1b-right**), which highlights a summary of model training across epochs, does not clearly demarcate convergence points for loss and performance. To better illustrate that the model has sufficiently converged by Epoch 50, we have added additional **Supplementary Figure 1F** (see attached below) and associated training log data with a narrow y-axis scale for both loss and correlation performance. We note that in the final four epochs (46 - 50), the delta loss falls below the pre-specified threshold defined in the EarlyStopping callback. Based additionally upon the convergence of accuracy score, as measured by R2, we determined empirically that further investment of resources into longer training would most likely not yield tangibly relevant increases in model performance. As shown in Figure 1E, we observed that increases in training dataset pseudo-mixture size demonstrated significant increases in model performance, which we modeled in Supplementary Figure 1A.

We have included a more detailed description of training procedures and stopping criteria in the revised **Methods** section “**Training Strategy**” (Lines 299 - 318).

Supplementary Figure 1F. Validation of Model Convergence. Re-scaled convergence plots for UCDBase training demonstrate performance convergence of both loss (left) and accuracy (right) metrics.

- It would be good if the authors could provide a supplementary table containing the list of 899 datasets.

We thank the Reviewer for the suggestion, and agree that providing the list of studies used for generating training data would be beneficial to readership. We have included an additional **Supplementary Table 5** containing all studies used to generate training data for the version of the model presented in the manuscript, coded by database accession number, PMID where available, publication date, and title. A reference to this list has been provided in the revised **Methods** section “**Quality Control and Manual Correction of Annotations**” (Line 293). An excerpt of this table is shown below:

accession	PMID	Date	Title
E-HCAD-18	30796046	2/24/19	Maturation Of Heart Valve Cell Populations During Postnatal Remodeling.
E-MTAB-10026	33879890	4/20/21	Deciphering the molecular immune response to COVID-19 using single cell multi-omics
E-MTAB-10197	29752062	5/13/18	Single-Cell Rna Sequencing Of Lymph Node Stromal Cells Reveals Niche-Associated Heterogeneity.
E-MTAB-2983	26444631	10/9/15	Adult Human And Mouse Ovaries Lack Ddx4-Expressing Functional Oogonial Stem Cells.

Reviewers' Comments:

Reviewer #1:

Remarks to the Author:

The authors have done a great job in revising the manuscript, particularly the newly added benchmarks. All my concerns are fully addressed, and the manuscript is much improved. I have no further comments.

Reviewer #2:

Remarks to the Author:

The authors have properly addressed my comments.

Reviewer #3:

Remarks to the Author:

Thanks to the authors for the nice clarifications and expansion of experiments provided in this revision. The major concerns about benchmarks are adequately addressed. Appreciate the additional feedback and no further comments on this study.

NCOMMS-22-32274A

Peer Review Comments & Responses

Introduction

We would like to extend our sincere gratitude to all the reviewers for providing their feedback on our revised manuscript. Given the supportive reviewer comments, we have addressed any outstanding editorial items as requested in the final revised manuscript. Overall we believe that the questions and recommendations provided throughout the review and editorial process have led to a substantial improvement in the method and manuscript as a whole.

Response Format

Reviewers' comments are addressed below. All **responses are denoted in green text.**

Reviewer 1

Original Remarks to the Author

The authors have done a great job in revising the manuscript, particularly the newly added benchmarks. All my concerns are fully addressed, and the manuscript is much improved. I have no further comments.

Overall Response

We thank Reviewer 1 for their encouraging comments surrounding the revisions made to the manuscript and for their insight throughout the review process.

Reviewer 2

Original Remarks to the Author

The authors have properly addressed my comments.

Overall Response

We thank Reviewer 2 for their positive comment and their feedback in the review process as a whole.

Reviewer 3

Original Remarks to the Author

Thanks to the authors for the nice clarifications and expansion of experiments provided in this revision. The major concerns about benchmarks are adequately addressed. Appreciate the additional feedback and no further comments on this study.

Overall Response

We thank Reviewer 3 for their supportive comments concerning the revisions made to the manuscript and their suggestions during the review process.